# Revisiting LARS for Large Batch Training Generalization of Neural Networks

## Abstract

LARS and LAMB have emerged as prominent techniques in Large Batch Learning (LBL) to ensure training stability in AI. Convergence stability is a challenge in LBL, where the AI agent usually gets trapped in the sharp minimizer. To address this challenge, warm-up is an efficient technique, but it lacks a strong theoretical foundation. Specifically, the warm-up process often reduces gradients in the early phase, inadvertently preventing the agent from escaping the sharp minimizer early on. In light of this situation, we conduct empirical experiments to analyze the behaviors of LARS and LAMB with and without a warm-up strategy. Our analyses give a comprehensive insight into the behaviors of LARS, LAMB, and the necessity of a warm-up technique in LBL, including an explanation of their failure in many cases. Building upon these insights, we propose a novel algorithm called Time Varying LARS (TVLARS), which facilitates robust training in the initial phase without the need for warm-up. A configurable sigmoid-like function is employed in TVLARS to replace the warm-up process to enhance training stability. Moreover, TVLARS stimulates gradient exploration in the early phase, thus allowing it to surpass the sharp optimizer early on and gradually transition to LARS and achieving robustness of LARS in the latter phases. Extensive experimental evaluations reveal that TVLARS consistently outperforms LARS and LAMB in most cases, with improvements of up to $2\%$ in classification scenarios. Notably, in every case of self-supervised learning, TVLARS dominates LARS and LAMB with performance improvements of up to $10\%$.

## 1 Introduction

Large Batch Learning (LBL) is crucial in modern Deep Learning (DL) for its efficiency gains through parallel processing, enhanced generalization with exposure to diverse samples, memory efficiency, and hardware utilization. These advantages make LBL particularly suitable for training large Deep Neural Network (DNN) models and Self-Supervised Learning (SSL) tasks, where increased model capacity and representation learning are crucial. Nonetheless, the present application of LBL with conventional gradient-based methods often necessitates the use of heuristic tactics and results in compromised generalization accuracy Hoffer et al. (2017); Keskar et al. (2017).

Numerous methods Huo et al. (2021); You et al. (2020); Fong et al. (2020) have been explored to address the performance issues associated with large-batch training. Among these methods, Layer-wise Adaptive Rate Scaling (LARS) You et al. (2017) has gained significant popularity. Fundamentally, LARS employs adaptive rate scaling to improve gradient descent on a per-layer basis. As a result, training stability is enhanced across the layers of the DNN model. Despite its benefits, LARS faces instability in the initial stages of the LBL process, leading to slow convergence, especially with large batches. Implementing a warm-up strategy is effective in reducing the LARS adaptive rate and stabilizing the learning process for larger batch sizes. However, this approach relies on a vague tuning process and lacks a solid theoretical foundation, providing opportunities for further exploration and improvement in adaptive rate scaling algorithms.

Through empirical experiments, we made an interesting observation in the initial phase of LARS training. The Layer-wise Learning Rate (LLR) was found to be high due to the Layer-wise Normalization Rate (LNR), i.e., $\|w\|/\|\nabla\mathcal{L}\|$, which was caused by the near-zero value of the Layer-wise Gradient Norm (LGN), i.e., $\|\nabla\mathcal{L}\|$. This infinitesimal value of LGN was a consequence of getting trapped into sharp minimizers (i.e., characterized by large positive eigenvalue of Hessian Keskar

et al. (2017); Dinh et al. (2017)) during the initial phase, leading to an explosion of the scaled gradient. When incorporating the current warm-up technique into LARS, it takes considerable unnecessary steps to scale the gradient to a threshold that enables escape from the initial sharp minimizers (see Figure 1). Furthermore, because of the constant decay in the Learning Rate (LR), the warm-up process does not effectively encourage gradient exploration over the initial sharp minimizers and struggles to adapt to diverse datasets. To address these issues, we propose a new algorithm called Time-Varying Layer-wise Adaptive Rate Scaling (TVLARS), which enables gradient exploration for LARS in the initial phase while retaining the stability of other LARS family members in the latter phase. Instead of using warm-up, which, as we will discuss later, suffers from major aforementioned issues, TVLARS, in contrast, overcomes sharp minimizes by taking full advantage of a high initial LR (i.e., target LR) in warm-up aided LARS (WA-LARS) and inverted sigmoid function enhancing training stability, aligning with theories about sharp minimizers in LBL Keskar et al. (2017). Our contributions can be summarized as follows:

- We perform empirical experiments on two canonical LBL techniques, namely LARS and LAMB, to gain a comprehensive understanding of how they enhance LBL performance.
- We investigated the necessity of warm-up for the LARS method and identified potential issues with using warm-up in LARS. These potential shortcomings arise from the lack of understanding regarding sharp minimizers aspects in LBL.
- Acknowledging the principles of LARS and the warm-up technique, we propose a simple and straightforward alternative technique, the so-called TVLARS, which is more aligned with the theories about sharp minimizers in LBL and can avoid the potential issues of the warm-up approach.
- To validate the efficacy of TVLARS, we conduct several experimental evaluations, comparing its performance against other popular baselines. The results of our experiments demonstrate that under the same delay step and target LR, TVLARS significantly outperforms the state-of-the-art benchmarks, especially WA-LARS particularly when the batch size becomes large (e.g., 8192, 16384).

## 2 NOTATIONS AND PRELIMINARIES

**Notation.** We denote by $w_t \in \mathbb{R}^d$ the model parameters at time step $t$. For any function $f : \mathbb{R}^d \to \mathbb{R}^d$, $\nabla l(x_i, y_i | w^k)$ is denoted the gradient with respect to $w^{(k)}$. We use $\| \cdot \|$ and $\| \cdot \|_1$ to denote $l_2$-norm and $l_1$-norm of a vector, respectively. We start our discussion by formally stating the problem setup. In this paper, we study a non-convex stochastic optimization problem of the form:

$$\min_{w \in \mathbb{R}^d} \mathcal{L}(w) \triangleq \mathbb{E}_{x_i, y_i \sim P(X,Y)}[\ell(x_i, y_i | w)] + \frac{\lambda}{2} \|w\|^2, \tag{1}$$

where $\ell$ is an empirical loss function, $(x, y) \sim P(X, Y) \in Z = \{X, Y\}$ is sample and ground truth.

**LARS.** To deal with LBL, You et al. (2017) proposed LARS. Suppose a neural network has $K$ layers, we have $w = \{w^1, w^2, \ldots, w^K\}$. The LR at layer $k$ is updated as follows:

$$\gamma_t^k = \gamma_{\text{scale}} \times \eta \times \frac{\|w_t^k\|_2}{\| \frac{1}{B} \sum_{i \in I_t} \nabla l(x_i, y_i | w_t^k) \|}, \tag{2}$$

where $\gamma_{\text{scale}} = \gamma_{\text{tuning}} \times \frac{\mathcal{B}}{\mathcal{B}_{\text{base}}}$ is the base LR You et al. (2017) and $\eta$ is the LARS coefficient for the LBL algorithm. We denote $\| \frac{1}{B} \sum_{i \in I_t} \nabla l(x_i, y_i | w_t^k) \|$ as the LGN. For simplicity, in later parts, we denote the LGN as $\|\nabla w\|$. The LNR is defined as $\frac{\|w_t^k\|_2}{\| \frac{1}{B} \sum_{i \in I_t} \nabla l(x_i, y_i | w_t^k) \|_2}$, which has the objective of normalizing the LR to each layer $k$. Despite its practical effectiveness, there is inadequate theoretical insight into LARS. Additionally, without implementing warm-up techniques Goyal et al. (2018), LARS tends to exhibit slow convergence or divergence during initial training.

## 3 THE MYSTERY OF GENERALIZATION BEHIND WARM-UP LARS AND NON WARM-UP LARS

This section explores how the LARS optimizers contribute to large-batch training. By revealing the mechanism of large-batch training, we provide some insights for improving LARS performance.

**Algorithm 1** TVLARS algorithm

---

**Require:** $w_t^k \in \mathbb{R}^d$, LR $\{\gamma_t^k\}_t^T$, delay factor $\lambda$, batch size $\mathcal{B}$, delay epoch $d_e$, scaling factor $\alpha$, time-varying factor $\phi_t$, momentum $\mu$, $\eta$, $\gamma_{\min}$,
 **for** $e = 1 : N$ **do**
  Sample $\mathcal{B}$ samples $\mathbb{B}_t = \left\{(x_t^1, y_t^1), \cdots, (x_t^b, y_t^b)\right\}$
  Compute gradient

$$\nabla_w^t \mathcal{L}(w) = \frac{1}{|\mathbb{B}_t|} \sum_{i=1}^b \nabla \ell(x_t^i, y_t^i | w).$$

  Update $\phi_t = \frac{1}{\alpha + e^{\psi_t}} + \gamma_{\min}$ where $\psi_t = \lambda(t - d_e)$
as mentioned in (4) and $\gamma_{\min}$ defined in (5).
  Compute layer-wise LR $\gamma_t^k$

$$\gamma_t^k = \eta \times \phi_t \times \frac{\|w_t^k\|}{\|\nabla_w^t \mathcal{L}(w) + w_d\|}$$

  Compute momentum $m_{t+1}^k = w_t^k - \gamma_t^k \nabla_w^t \mathcal{L}(w)$
  Adjust model weight

$$w_{t+1}^k = m_{t+1}^k + \mu \left(m_{t+1}^k - m_t^k\right).$$

**end for**

---

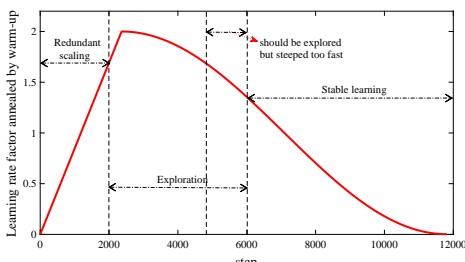

(a) WA-LARS.

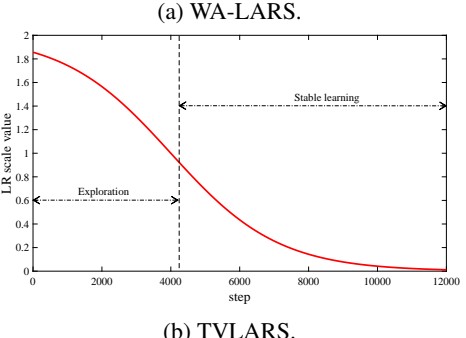

(b) TVLARS.

Figure 1: Scaling value of the learning rate in two different strategies.

To understand the detrimental impact of lacking warm-up procedure in current state-of-the-art LBL techniques, we employ the LARS and LAMB optimizers on vanilla classification problems to observe the convergence behavior. We conduct empirical experiments on CIFAR10 Krizhevsky (2009).

### 3.1 On the principle of LARS

First, we revisit the LARS algorithm, which proposes adaptive rate scaling. Essentially, LARS provides a set of LR that adapts prior to each layer of the DNN, as shown in Equation (2). From a geometric perspective on layer $k$ of the DNN, the layer-wise weight norm (LWN) $\|w_t^k\|$ can be seen as the magnitude of the vector containing all components in the Euclidean vector space. Similarly, the LGN can be regarded as the magnitude of the gradient vector of all components in the vector space. Thus, the LNR can be interpreted as the number of distinct pulses in Hartley's law.

By considering the LNR, we can adjust the layer-wise gradient based on the LWN. In other words, instead of taking the normal gradient step $\nabla w_t^k$ at every layer $k$, we perform a gradient step as a percentage of the LWN. The proportional gradient update can be expressed as follows:

$$\gamma_{\text{scale}} \times \eta \times \frac{\|w_t^k\|}{\|\nabla \mathcal{L}(w_t^k)\|} \times \nabla \mathcal{L}(w_t^{k,j}) = \gamma_{\text{scale}} \times \eta \times \|w_t^k\| \times \frac{\nabla \mathcal{L}(w_t^{k,j})}{\|\nabla \mathcal{L}(w_t^k)\|}, \quad (3)$$

where $\nabla \mathcal{L}(w_t^{k,j})$ is the gradient on $j$-th parameter in layer $k$. $\frac{\nabla \mathcal{L}(w_t^{k,j})}{\|\nabla \mathcal{L}(w_t^k)\|}$ represents a function that estimates the percentage of gradient magnitude on each parameter $j$ with respect to the LGN of layer $k$. It becomes apparent that the LLR function of LARS only influences the percentage update to the layer-wise model parameters. However, it does not address the issue of mitigating the problem of sharp minimizers in the initial phase of LBL.

### 3.2 LARS and the importance of warm-up

The warm-up strategy is considered an important factor of LBL, which enhances the model performance and training stability You et al. (2020), Gotmare et al. (2019), Goyal et al. (2018), You et al. (2017). The warm-up strategy linearly scales the LR from 0 to the target one, then switches to a reg-

ular LR decay, which is stated to reduce loss in accuracy, though its unproven empirical properties. Therefore, we analyzed to investigate the vitality of warming up as well as its potential issues.

**Quantitative results.** Our quantitative results on CIFAR10 are presented in Appendix H, showcasing the accuracy of training on the test dataset. The results reveal a decline in AI performance when contrasting runs with and without a warm-up strategy. In particular, the LARS without a warm-up technique exhibits greater training instability, characterized by fluctuating accuracy. Moreover, the performance decline becomes more significant, especially with larger batch sizes.

**From adaptive ratio to LBL performance.** To enhance our comprehension of the adaptive rate scaling series, we conducted thorough experiments analyzing LNR in LARS. Each result in our study includes two crucial elements: the test loss during model training and the corresponding LNR. In our study, we examined the detailed results of WA-LARS, as presented in Appendix G.2.1. In

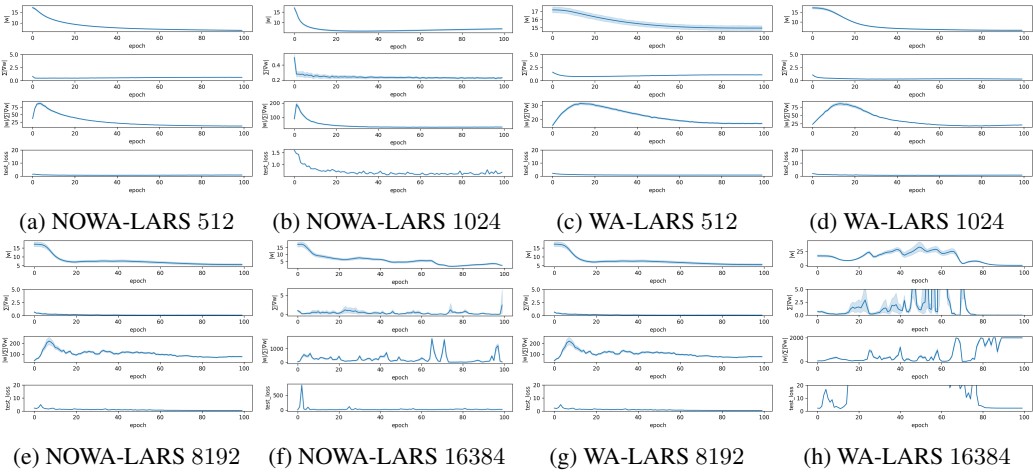

| (a) NOWA-LARS 512 | (b) NOWA-LARS 1024 | (c) WA-LARS 512 | (d) WA-LARS 1024 |
|---|---|---|---|

| (e) NOWA-LARS 8192 | (f) NOWA-LARS 16384 | (g) WA-LARS 8192 | (h) WA-LARS 16384 |
|---|---|---|---|

Figure 2: This figure illustrated the quantitative performance of LARS ($B = 16K$) conducted with a warm-up and without a warm-up strategy (NOWA-LARS). Each figure contains 4 subfigures, which indicate the LWN $\|w\|$, LGN $\|\nabla w\|$, and LNR $\|w\|/\|\nabla w\|$ of all layers, and test loss value in the y axis. The comprehensive version is shown in Appendix F.

addition, we made observations regarding the convergence and the behavior of the LNR:

**1)** During the initial phase of the successfully trained models (characterized by a significant reduction in test loss), the LNR tends to be high, indicating a higher LR.

**2)** High variance in the LNR indicates significant exploration during training, resulting in noticeable fluctuations in LWM. Conversely, when training requires stability, the LNR variance decreases.

**3)** Additionally, we noticed that it is necessary to impose an upper threshold on the LNR to prevent divergence caused by values exceeding the range of $\|\gamma_t^k \times \nabla \mathcal{L}\|$ over the LWN $\|w\|$ You et al. (2017). The absence of the warm-up technique in LARS often leads to the LNR surpassing this upper threshold. This issue is addressed by the WA-LARS. Specifically, compared with non warm-up LARS (NOWA-LARS) at batch sizes of $512, 1024, 2048, 4096, 8192$, respectively, where the LNR during the initial phase is limited to values such as $0.15, 0.2, 0.3, 0.5, 1.5$ (see Figures 20, 21, 22, 23, 24), the highest LNR of WA-LARS is $0.06, 0.1, 0.125, 0.15, 0.25$ (see Figures 14, 15, 16, 17, 18).

**4)** The LNR in the WA-LARS is regulated by a more gradual incline. For example, after $40 - 50$ epochs, the LNR decreases from $0.060$ to $0.018$ in Figure 14, $0.10$ to $0.02$, $0.125$ to $0.027$ in Figure 15, $0.16$ to $0.048$ in Figure 16, and $0.20$ to $0.050$ in Figure 17 for batch sizes of $512, 1024, 2048, 4096$, respectively. On the other hand, in contrast to the WA-LARS, the NOWA-LARS exhibits a steep decline in the LNR. For instance, after $10 - 20$ epochs, the LNR decreases from $0.15$ to $0.048$ in Figure 20, $0.221$ to $0.032$ in Figure 21, $0.30$ to $0.043$ in Figure 22, and $0.43$ to $0.089$ in Figure 23. For more comprehensive details and additional results, please refer to Appendix H.

**Extensive Study.** To gain deeper insights, we conducted additional experiments, and the results are presented in Figure 2. Through the analysis of Section 3.2 and the insights derived from Figure 2, it becomes evident that the reduction in the LNR can be attributed to the rapid decrease in the LWN over time. This phenomenon occurs in tandem with the exponential reduction of the LGN. Consequently, we can deduce that the superior performance of the AI model with respect to larger batch sizes is a consequence of the gradual decrease in the LWN.

Put differently, envisioning the model parameters as a hypersphere, the gradient descent technique explores the topological space of this hypersphere. This exploration commences from the hypersphere's edge, indicated by $\|w_t^k\| = w_{\max}$, and progresses toward its center, characterized by $\|w_t^k\| = 0$. This hypothesis finds support in the gradual decrease of $\|w_t^k\|$ as depicted in Figure 2. Nevertheless, in cases of exploding gradient issues, significant fluctuations in $\|w_t^k\|$ can disrupt the functionality of this hypothesis.

In the context of NOWA-LARS, the LNR experiences a steeper decline due to the rapid reduction in the LWN $\|w_t^k\|$. This high reduction rate can be understood as an overly swift exploration of the parameter vector hypersphere. Such expeditious exploration results in overlooking numerous potential searches, leading to the potential failure to identify global minimizers. In contrast, with WA-LARS, the search across the parameter space occurs more gradually, ensuring a more stable exploration of the parameter hypersphere.

Based on the aforementioned observations, our conclusion is that the primary challenges encountered in the context of LARS stem from two key issues: a high LNR and substantial variance in the LWN. These dual challenges pose significant obstacles to LARS's effective performance. As a solution, the warm-up process aims to prevent the occurrence of exploding gradients during the initial phase by initially setting the LR coefficient to a significantly low value and gradually increasing it thereafter. However, we believe that the application of the warm-up technique may be somewhat lacking in a comprehensive understanding. Consequently, we are motivated to delve deeper into the characteristics of sharp minimizers within the LBL, seeking a more profound insight into LARS and the warm-up process.

## 3.3 SHORTCOMINGS OF WARM-UP

**The degradation in learning performance when the batch size becomes large and the sharp minimizer.** As mentioned in Section 3.1, the LARS technique only influences the percentage update to their layer-wise model parameters to stabilize the gradient update behavior. However, the learning efficiency is not affected by the LARS technique. To gain a better understanding of LARS performance as the batch size increases significantly, our primary goal is to establish an upper limit for the unbiased gradient, which is similar to (Wang et al., 2020, Assumption 2) (i.e., the variance of the batch gradient). We first adopt the following definition:

**Definition 1.** *A gradient descent $g_i^t$ at time $t$ using reference data point $x_i$ is a composition of a general gradient $\bar{g}^t$ and a variance gradient $\Delta g_i^t$. For instance, we have $g_i^t = \bar{g}^t + \Delta g_i^t$, where the variance gradient $\Delta g_i^t$ represents the perturbation of gradient descent over the dataset. The general gradient represents the invariant characteristics over all perturbations of the dataset.*

The aforementioned definition leads to the following theorem that shows the relationship between unbiased gradient (Wang et al., 2020, Assumption 2) and the batch size as follows:

**Theorem 1** (Unbiased Large Batch Gradient)**.** *Given $\bar{g}^t$ as mentioned in Definition 1, $g_{\mathcal{B}}^t$ is the batch gradient with batch size $\mathcal{B}$. Given $\sigma^2$ is the variance for point-wise unbiased gradient as mentioned in (Wang et al., 2020, Assumption 2), we have the stochastic gradient with $\mathcal{B}$ is an unbiased estimator of the general gradient and has bounded variance: $\mathbb{E}_{x_i, y_i \in \{\mathcal{X}, \mathcal{Y}\}} [\bar{g}^t - g_{\mathcal{B}}^t] \leq \sigma^2/\mathcal{B}$,*

*Proof.* The proof is shown in Appendix C.

Theorem 1 demonstrates that utilizing a large batch size $\mathcal{B}$ during training results in more stable gradients. However, there are two significant concerns with this which come with negative implications. Firstly, the stability of the gradient is influenced by the large batch (LB). Consequently, in scenarios where the LNR experiences rapid reduction (as discussed in NOWA-LARS in Section 3.2), there is a potential for the gradient descent process to become trapped into sharp minimizers during the initial stages Keskar et al. (2017). Secondly, due to the steep decline in the LNR, the exploration across the hypersphere of $w_k^t$ occurs excessively swiftly (specifically, from $w_{\max}$ to 0). LB techniques lack the

exploratory characteristics of small batch (SB) methods and often focus excessively on narrowing down to the minimizer that is closest to the starting point Keskar et al. (2017).

**Redundant Ratio Scaling in Warm-up LARS.** Warm-up Gotmare et al. (2019) involves initially scaling the base LR and subsequently reducing it to facilitate gradient exploration. However, our findings indicate that gradually increasing the base LR from an extremely low value before gradient exploration is unnecessary (see Figure 1). As a consequence, when we multiply the base LR with the LNR (which tends to be low in a few initial rounds), the LLR will be extremely low accordingly. To verify this assumption, we conduct empirical experiments to see the LLR of each layer of DNN and demonstrate it as in Appendix J.

When the LLR $\gamma_t^k$ is too small, particularly at the initial stage, learning fails to avoid memorizing noisy data You et al. (2019). Moreover, when the model gets trapped in the sharp minimizers during the warm-up process, due to the steepness of the sharp minimizers, the model will be unable to escape from the sharp minima. Furthermore, apart from the high variance of the gradient of mini-batch learning, the gradient of the LBL is stable as mentioned in Theorem 1. Therefore, the LBL is halted until the LR surpasses a certain threshold.

## 4 METHODOLOGY

After conducting and comprehending the experiential quantification in Section 3, we have identified the issues of warm-up LARS. It becomes evident that the LLR is not well-aligned with the characteristics of sharp minimizer distributions in the context of LB settings. Specifically, during the initial phases of the search process, the loss landscape tends to exhibit numerous sharp minimizers that necessitate sufficiently high gradients to facilitate efficient exploration. Furthermore, it is imperative for the LR to be adjustable so that the LBL can be fine-tuned to match the behavior of different datasets and learning models.

To this end, we propose a novel algorithm TVLARS for LBL optimization that aims to take full advantage of the strength of LARS and warm-up strategy along with drawbacks avoidance. A key idea of TVLARS is to ensure the following characteristics: 1) elimination of incremental phase of base LR to eliminate redundant unlearnable processes, 2) a configurable base LR function that can be tuned for different data and model types, and 3) a lower threshold for stability and inheriting LARS robustness.

**1) Initiating Exploration Stimulus.** Although warm-up strategy enhances model training stability (Section 3), learning from a strictly small LR prevents the model from tackling poor sharp minimizers, appearing much near initialization point Granziol et al. (2022), Keskar et al. (2017), Dinh et al. (2017). Otherwise, as a result of the steep decline in adaptive LNR, the exploration around the hypersphere of $w_k^t$ is restricted (Theorem 1) and does not address the sharp minimizer concern (Section 3.1). Moreover, warm-up does not fulfill the need for LBL training because of the unnecessary linear scaling (Section 3.3). We construct TVLARS as an optimizer that uses a high initial LR owing to its ability of early sharp minima evasion, which enhances the loss landscape exploration.

$$\phi_t = \frac{1}{\alpha + e^{\psi_t}} + \gamma_{\min} \quad \text{where} \quad \psi_t = \lambda(t - d_{\mathrm{e}}) \tag{4}$$

**2) Learning Rate Decay.** To avoid instability of training due to the high initial LR, we use a parameter $d_{\mathrm{e}}$ specifying the number of delayed epochs as inspired by the warm-up strategy. After $d_{\mathrm{e}}$ epochs, the base LR is annealed via Equation (4), which is the time-varying component used to tackle. According to the mathematical discussion of the LARS principle (Section 3.1) and the aforementioned characteristic of LARS, LAMB with and without a warm-up strategy (Section 3.2, 3.3), the LNR $\frac{\|w\|}{\|\nabla\mathcal{L}(w)\|}$ tends to be exploding as the model is stuck at local sharp minima, then $\|\nabla\mathcal{L}(w)\|$ decay faster than $\|w\|$.

**3) Configurable Decay Rate.** The proposed time-varying component $\phi_t$ is constructed based on the sigmoid function whose curve is smooth to keep the model away from unstable learning (see Figure 3). $\lambda$ is the soft-temperature factor, which controls the steepness of the time-variant component. Specifically, as $\lambda$ is large, the steepness is significant, and the time-variant component $\phi_t$ reduces faster. Therefore, by changing the soft-temperature factor $\lambda$, we can adjust the transition duration from gradient exploration to stable learning (i.e., we can achieve a stable learning phase faster as $\lambda$ is larger, and otherwise).

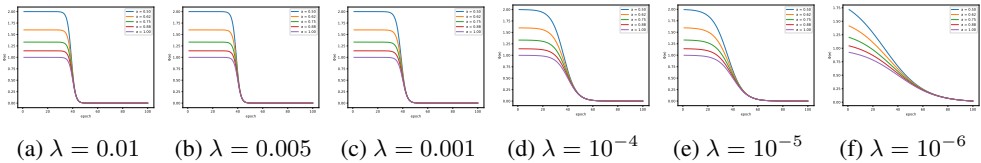

| (a) $\lambda = 0.01$ | (b) $\lambda = 0.005$ | (c) $\lambda = 0.001$ | (d) $\lambda = 10^{-4}$ | (e) $\lambda = 10^{-5}$ | (f) $\lambda = 10^{-6}$ |

Figure 3: The decay plot of TVLARS algorithm under different settings.

**4) Alignment with LARS.** When the learning process is at the latter phase, it is essential for the TVLARS behavior to align with that of LARS in order to inherit LARS's robustness (see Figure 3). We introduce two parameters $\alpha$ and $\gamma_{\min}$ used to control the bound for time-varying component $\phi_t$. For any $\alpha, \gamma_{\min} \in \mathbb{R}$, the lower and upper bounds for $\phi_t$ is shown in Equation (5) (see proof in Appendix D).

$$\gamma_{\min} \leq \phi_t \leq \frac{1}{\alpha + \exp\{-\lambda d_e\}} \tag{5}$$

This boundary ensures that the LLR does not explode during the training phase. Otherwise, to guarantee that all experiments conducted are fair among optimizers, $\alpha$ is set to 1, which means there is no increment in the initial LR after being chosen, and the minimum value of the LR is also set to $\gamma_{\min}$.

## 5 EXPERIMENT

Table 1: Accuracy (%) of LARS, LAMB, and TVLARS on a classification ($\lambda = 10^{-4}$), and SSL ($\lambda = 10^{-5}$). Weight initialization is Xavier Uniform

| Problem | | Classification | | | | | | SSL - Barlow Twins | | | | | |
|---|---|---|---|---|---|---|---|---|---|---|---|---|---|
| Data set | | CIFAR10 | | | Tiny ImageNet | | | CIFAR10 | | | Tiny ImageNet | | |
| Learning rate | | 1 | 2 | 3 | 1 | 2 | 3 | 1 | 2 | 3 | 1 | 2 | 3 |
| LARS | | 74.64 | 77.49 | 79.64 | 33.72 | 36.60 | 38.92 | 51.31 | 58.89 | 60.76 | 18.32 | 19.52 | 21.12 |
| LAMB | $B = 512$ | 57.88 | 64.76 | 78.39 | 20.28 | 41.40 | 37.52 | 10.01 | 12.01 | 67.31 | 13.46 | 17.75 | 27.37 |
| TVLARS | | **78.92** | **81.42** | **81.56** | **37.56** | **39.28** | **39.64** | **69.96** | **69.72** | **70.54** | **29.02** | **31.01** | **31.44** |
| Learning rate | | 2 | 3 | 4 | 2 | 3 | 4 | 2 | 3 | 4 | 2 | 3 | 4 |
| LARS | | 74.70 | 78.83 | 80.52 | 33.8 | 36.84 | 38.52 | 61.13 | 67.03 | 68.98 | 19.96 | 19.36 | 20.38 |
| LAMB | $B = 1024$ | 52.06 | 57.83 | 79.98 | 16.88 | 31.84 | 37.76 | 12.03 | 15.03 | 69.49 | 16.44 | 16.70 | 24.53 |
| TVLARS | | **81.84** | **82.58** | **82.54** | **39.60** | **39.16** | **39.40** | **67.38** | **69.80** | **71.13** | **28.98** | **27.46** | **28.32** |
| Learning rate | | 5 | 6 | 7 | 5 | 6 | 7 | 5 | 6 | 7 | 5 | 6 | 7 |
| LARS | | 75.22 | 79.49 | 81.1 | 34.48 | 38.04 | 40.44 | 52.42 | 57.35 | 52.42 | 19.92 | 20.33 | 19.98 |
| LAMB | $B = 2048$ | 43.12 | 47.88 | 81.43 | 12.60 | 18.56 | 39.92 | 10.01 | 15.64 | 60.08 | 18.67 | 21.98 | 23.43 |
| TVLARS | | **81.52** | **82.22** | **82.44** | **39.56** | **39.56** | **41.68** | **62.61** | **61.71** | **61.05** | **26.15** | 23.43 | **27.08** |
| Learning rate | | 8 | 9 | 10 | 8 | 9 | 10 | 8 | 9 | 10 | 8 | 9 | 10 |
| LARS | | 75.63 | 80.96 | 82.49 | 34.24 | 38.56 | 40.40 | 52.77 | 55.74 | 55.98 | 19.45 | 20.20 | 20.25 |
| LAMB | $B = 4096$ | 22.32 | 37.52 | 81.53 | 05.24 | 10.76 | 39.00 | 48.93 | 50.19 | 53.38 | 13.78 | 15.66 | 21.45 |
| TVLARS | | **80.9** | 80.96 | 81.16 | **38.28** | **39.96** | **41.64** | **57.96** | **58.28** | **60.46** | **24.29** | **24.29** | **24.92** |
| Learning rate | | 10 | 12 | 15 | 10 | 12 | 15 | 10 | 12 | 15 | 10 | 12 | 15 |
| LARS | | 77.59 | 81.75 | 82.5 | 34.36 | 38.12 | 42.00 | 50.29 | 52.78 | 09.65 | 19.69 | 20.79 | 21.62 |
| LAMB | $B = 8192$ | 16.14 | 19.85 | 81.78 | 01.12 | 02.48 | 39.20 | 42.22 | 45.26 | 52.39 | 19.60 | 23.44 | 23.55 |
| TVLARS | | **81.16** | **82.42** | **82.74** | **36.48** | **39.20** | 42.32 | **54.14** | **53.56** | 52.24 | **23.41** | **23.47** | **24.26** |
| Learning rate | | 15 | 17 | 19 | 15 | 17 | 19 | 15 | 17 | 19 | 15 | 17 | 19 |
| LARS | | 79.46 | 80.62 | 38.57 | 35.60 | 39.72 | 41.72 | 48.27 | 48.26 | 49.05 | 20.97 | 21.54 | 22.42 |
| LAMB | $B = 16384$ | 14.82 | 11.97 | 77.16 | 00.84 | 00.92 | 37.28 | 42.26 | 44.02 | 49.65 | 20.34 | 23.58 | 23.78 |
| TVLARS | | 79.20 | 80.42 | **80.42** | **38.56** | **40.32** | 41.08 | 48.16 | **49.84** | **50.15** | **25.82** | **24.95** | **25.91** |

### 5.1 CLASSIFICATION AND BARLOW TWINS PROBLEM

Table 1 demonstrates the model performance trained with LARS, LAMB, and TVLARS (detail setting at B). The table contains two main columns for CIFAR and Tiny ImageNet with associated tasks: CLF and BT. Besides CIFAR, Tiny ImageNet is considered to be a rigorously challenging dataset (100, 000 images of 200 classes), which is usually used to evaluate the performance of LBL SSL tasks. Overall, TVLARS achieves the highest accuracies, which outperforms LARS $4 \sim 7\%$,

$3 \sim 4\%$, and $1 \sim 2\%$ in each pair of $\gamma_{\text{target}}$ and $\mathcal{B}$. LARS and LAMB, besides, are immobilized by the poor sharp minima indicated by the LNR $\|w\|/\|\nabla_w \mathcal{L}(w)\| \to \infty$ as $\|\nabla_w \mathcal{L}(w)\| \to 0$ (see Figure 2), which although creates a high adaptive LR to escape the trapped minima, $\nabla_w \mathcal{L}(w)/\|\nabla_w \mathcal{L}(w)\|$ only influences the percentage update to the layer-wise model parameters hence cannot tackle the problem of sharp minima thoroughly (more analysis at 3, E, G). This phenomenon is caused by the warm-up strategy partly making LARS and LAMB converge slowly and be stuck at sharp minima. TVLARS ($\lambda = 10^{-3}$) reach the optimum after 20 epochs, compared to $60 \sim 80$ epochs from LARS and LAMB ($\gamma_{\text{target}} = 19, \mathcal{B} = 16K$). See more results at E, G.1, G.2.

## 5.2  ABLATION TEST

### 5.2.1  DECAY COEFFICIENTS

Decay coefficient $\lambda$ is a simple regularized parameter, used to anneal the LR to enhance the model performance. Figure 4 demonstrates the experiments' result ($\mathcal{B} \in \{1024, 16384\}$) conducted with values of $\lambda$ (E, G.3, I). Otherwise, we set $\alpha = 1$, so that the $\gamma_{\text{target}}$ for all experiments are the same. Besides, $d_e$, the number of delay epochs is set to 10 and $\gamma_{\min}$ is set to $\frac{\mathcal{B}}{\mathcal{B}_{\text{base}}} \times 0.001$ for both TVLARS and LARS experiment.

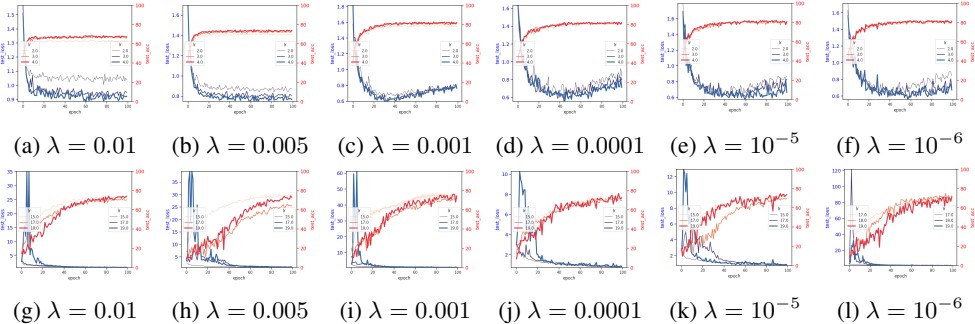

(a) $\lambda = 0.01$  (b) $\lambda = 0.005$  (c) $\lambda = 0.001$  (d) $\lambda = 0.0001$  (e) $\lambda = 10^{-5}$  (f) $\lambda = 10^{-6}$

(g) $\lambda = 0.01$  (h) $\lambda = 0.005$  (i) $\lambda = 0.001$  (j) $\lambda = 0.0001$  (k) $\lambda = 10^{-5}$  (l) $\lambda = 10^{-6}$

Figure 4: Quantitative comparison in learning stability between experiments ($\mathcal{B} \in \{1024, 16384\}$, which are upper and lower row, respectively). The bigger version is shown in Appendix I.

In 1K batch-sized experiments, there is a large generalization gap among $\gamma_{\text{target}}$ for $\lambda \in \{0.01, 0.005\}$. Smaller $\lambda$, otherwise, enhance the model accuracy by leaving $\gamma_{\text{target}}$ to stay nearly unchanged longer, which boosts the ability to explore loss landscape and avoid sharp minima. As a result, the model achieve higher accuracy: $\sim 84\%$ ($\lambda = 10^{-5}$), compared to result stated in 1 ($\lambda = 0.0001$). See E, I for detailed analysis and discussion. In contrast, the model performs better with larger values of $\lambda$ ($\lambda \in \{0.01, 0.005, 0.001\}$) in 16K batch-sized experiments. Owing to the pretty high initial $\gamma_{\text{target}}$, the model tends to find it easier to escape the local minima which do not converge within 20 epochs (four times compared to LARS) but also to a low loss value ($\sim 2$), compared to just under 20 ($\lambda = 10^{-6}$). This problem is owing to the elevated $\gamma_{\text{target}}$, which makes the leaning direction fluctuate dramatically in the latter training phase (see Figures 4j, 4k, 4l).

### 5.2.2  LEARNING RATE

A high initial LR, otherwise, plays a pivotal role in enhancing the model performance by sharp minimizer avoidance Keskar et al. (2017); Dinh et al. (2017). Authors of Granziol et al. (2022); Krizhevsky (2014) suggest that, when $\mathcal{B}/\mathcal{B}_{\text{base}} = m$, the LR should be $\epsilon \sqrt{m}$ to keep the variance, where $\epsilon$ is the LR used with $\mathcal{B}_{\text{base}}$. However, choosing $\epsilon$ is an empirical task, hence we do not only apply the theorem from Keskar et al. (2017) but also conduct the experiments with LRs in a large variation to analyze how LR can affect the model performance. Figure 5 illustrates that the higher the LR is, the lower the loss value and the higher the accuracy model can achieve (see detail analysis at E, G).

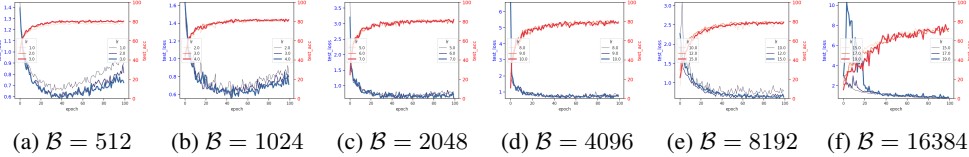

| (a) $\mathcal{B} = 512$ | (b) $\mathcal{B} = 1024$ | (c) $\mathcal{B} = 2048$ | (d) $\mathcal{B} = 4096$ | (e) $\mathcal{B} = 8192$ | (f) $\mathcal{B} = 16384$ |
|---|---|---|---|---|---|

Figure 5: Quantitative analysis of $\gamma_{\text{target}}$ ($\lambda = 0.0001$). Bigger version at Appendix I.

### 5.2.3 WEIGHT INITIALIZATION

According to You et al. (2017), the weight initialization is sensitive to the initial training phase. From Equation (2), when the value of $\gamma_t^k$ is high due to the ratio $\mathcal{B}/\mathcal{B}_{\text{base}}$ (i.e. $\mathcal{B} = 16K$), the update magnitude of $\|\gamma_t^k \nabla \mathcal{L}(w_t^k)\|$ may outperform $\|w_t^k\|$ and cause divergence. Otherwise, since $w \sim \mathcal{P}(w)$, which makes $\|w\|$ varies in distinguished variation, hence the ratio LNR $\|w\|/\|\nabla \mathcal{L}\|$ may make the initial training phase performance different in each method of weight initialization. Addressing this potential phenomenon, apart from Xavier Uniform Glorot & Bengio (2010), which has been shown above, we conducted the experiments using various types of weight initialization: Xavier Normal Glorot & Bengio (2010) and Kaiming He Uniform, Normal He et al. (2015b). It is transparent that, the model performance results using different weight initialization methods are nearly unchanged. TVLARS, though its performance is unstable owing to its exploration ability, outperforms LARS $1 \sim 3\%$ in both CIFAR10 and Tiny ImageNet.

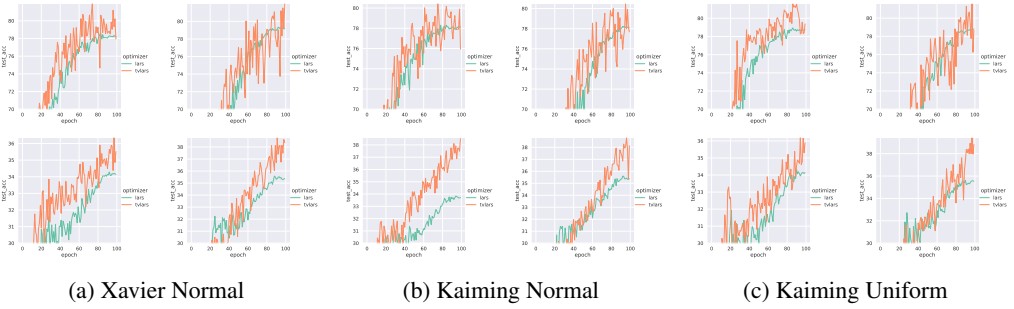

| (a) Xavier Normal | (b) Kaiming Normal | (c) Kaiming Uniform |
|---|---|---|

Figure 6: Quantitative analysis of different weight initialization methods for CIFAR10 and TinyImageNet (upper and lower rows). For each method, $\mathcal{B} \in \{8192, 16384\}$ (left, right columns)
.

## 6 CONCLUSION

Given the current lack of clarity in the fine-tuning and enhancing of the performance of layerwise adaptive LRs in LBL, we conducted extensive experiments to gain deeper insights into layerwise-based update algorithms. We then developed a new algorithm, referred to as TVLARS, based on our understanding of LARS and LAMB and their interaction with LBL. Our TVLARS approach capitalizes on the observation that LBL often encounters sharper minimizers during the initial stages. By prioritizing gradient exploration, we facilitate more efficient navigation through these initial obstacles in LBL. Simultaneously, through adjustable discounts in layerwise LRs, TVLARS combines the favorable aspects of a sequence of layerwise adaptive LRs to ensure strong convergence in LBL and overcome the issues of warm-up. With TVLARS we achieve significantly improved convergence compared to two other cutting-edge methods, LARS and LAMB, especially combined with warm-up and when dealing with extremely large batch sizes (e.g., $\mathcal{B} = 16384$), across Tiny ImageNet and CIFAR-10 datasets.

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

## A  RELATED WORKS

**Large-batch learning.** In Codreanu et al. (2017), several LR schedulers are proposed to figure out problems in LBL, especially the Polynomial Decay technique which helps ResNet50 converge within 28 minutes by decreasing the LR to its original value over several training iterations. Since schedulers are proven to be useful in LBL, You et al. (2017), Peng et al. (2018) suggested a LR scheduler based on the accumulative steps and a GPU cross Batch Normalization. Gotmare et al. (2019), besides, investigates deeper into the behavior of cosine annealing and warm-up strategy then shows that the latent knowledge shared by the teacher in knowledge distillation is primarily disbursed in the deeper layers. Inheriting the previous research, Goyal et al. (2018) proposed a hyperparameter-free linear scaling rule used for LR adjustment by constructing a relationship between LR and batch size as a function.

**Adaptive optimizer.** Another orientation is optimization improvement, starting by You et al. (2017), which proposed LARS optimizers that adaptively adjust the LR for each layer based on the local region. To improve LARS performance, Jia et al. (2018) proposed two training strategies including low-precision computation and mixed-precision training. In contrast, Liu & Mozafari (2022) authors propose JointSpar and JointSpar-LARS to reduce the computation and communication costs. On the other hand, Accelerated SGD Yamazaki et al. (2019), is proposed for training DNN in large-scale scenarios. In You et al. (2020), Xu et al. (2023), new optimizers called LAMB and SLAMB were proved to be successful in training Attention Mechanisms along with the convergence analysis of LAMB and LARS. With the same objective AGVM Xue et al. (2022) is proposed to boost RCNN training efficiency. Authors in Fong et al. (2020), otherwise proposed a variant of LAMB called LAMBC which employs trust ratio clipping to stabilize its magnitude and prevent extreme values. CLARS Huo et al. (2021), otherwise is suggested to exchange the traditional warm-up strategy owing to its unknown theory.

## B  EXPERIMENTAL SETTINGS

**Problems.** The vanilla classification (CLF) and Self Supervised Learning (SSL) are conducted and evaluated by the accuracy (%) metric. Regarding the success of SSL, we conduct the SOTA Barlow Twins[1] (BT) Zbontar et al. (2021) to compare the performance between LARS, LAMBYou et al. (2020), and TVLARS (ours). To be more specific, the BT SSL problem consists of two stages: SSL and CLF stage, conducted with 1000 and 100 epochs, respectively. The dimension space used in the first stage of BT is 4096 stated to be the best performance setting in Zbontar et al. (2021), along with two sub Fully Connected 2048 nodes layers integrated before the latent space layer. We also conduct the CLF stage of BT with vanilla Stochastic Gradient Descent (SGD) Kiefer & Wolfowitz (1952) along with the Cosine Annealing Loshchilov & Hutter (2017) scheduler as implemented by BT authors. The main results of CLF and BT tasks are shown at 5.1. See more results of LAMBYou et al. (2020), LARSYou et al. (2017), and TVLARS at G.1, G.2, G.3, separately.

**Datasets and Models.** To validate the performance of the optimizers, we consider two different data sets with distinct contexts: CIFAR10 Krizhevsky (2009) ($32 \times 32$, 10 modalities) and Tiny ImageNet Deng et al. (2009) ($64 \times 64$, 200 modalities). Otherwise, the two SOTA model architectures ResNet18 and ResNet34 He et al. (2016) are trained separately from scratch on CIFAR10 and TinyImageNet. To make a fair comparison between optimizers, the model weight is initialized in Kaiming Uniform Distribution He et al. (2015a).

**Optimizers and Warm-up Strategy.** Specifically, we explore the characteristics of LARS and LAMB by applying them with and without a warm-up strategy, aiming to understand the LNR $|w|/|\nabla\mathcal{L}(w)|$ (see more results at G.2, H. LARS and LAMB official source codes are implemented inside NVCaffe Jia et al. (2014) and Tensorflow Abadi et al. (2016). the Pytorch version of LARS used in this research is verified and referenced from Lightning Flash [2]. LAMB Pytorch code, on the other hand, verified and referenced from Pytorch Optimizer [3]. Besides, the warm-up strategy Gotmare et al. (2019) contains two separate stages: linear LR scaling and LR decay. In this first stage, $\gamma_t$ becomes greater gradually by iteratively updating $\gamma_t = \gamma_{\text{target}} \times \frac{t}{T}$ for each step ($T = 20$

---

[1]https://github.com/facebookresearch/barlowtwins
[2]https://github.com/Lightning-Universe/lightning-flash
[3]https://github.com/jettify/pytorch-optimizer

epochs). Then, $\gamma_t$ goes down moderately by $\gamma_t = \gamma_{\text{target}} \times q + \gamma_{min} \times (1 - q)$ where $q = \frac{1}{2} \times (1 + \cos \frac{\pi t}{T})$, which is also conducted in Zbontar et al. (2021); Chen et al. (2020); Bardes et al. (2022). In experiments where LARS and LAMB are conducted without a warm-up strategy, a simple Polynomial Decay is applied instead. TVLARS, on the contrary, is conducted without using a LR scheduler.

**Hyperparameters and System.** The LRs are determined using the square root scaling rule Krizhevsky (2014), which is described detailedly at 5.2.2 (see more results at I). We consider the following sets of $\gamma_{\text{target}}$: $\{1, 2, 3\}$, $\{2, 3, 4\}$, $\{5, 6, 7\}$, $\{8, 9, 10\}$, $\{10, 12, 15\}$, and $\{15, 17, 19\}$, which are associated with $\mathcal{B}$ of 512, 1024, 2048, 4096, 8192, and 16384, respectively. Otherwise, $w_d$, and $\mu$ is set to $5 \times 10^{-4}$, and 0.9, respectively. Besides, all experiments are conducted on Ubuntu 18.04 by using Pytorch Paszke et al. (2019) with multi Geforce 3080 GPUs settings, along with Syncing Batch Normalization, which is proven to boost the training performance Krizhevsky (2014), Yao et al. (2021), Huang et al. (2018).

## C   PROOF ON THEOREM 1

Revisit the Definition 1, we have:

$$g_i^t = \bar{g}^t + \Delta g_i^t. \tag{6}$$

In applying the LB gradient descent with batch size $\mathcal{B}$, we have:

$$g_{\mathcal{B}}^t = \frac{1}{\mathcal{B}} \sum_{i=1}^{\mathcal{B}} g_i^t = \frac{1}{\mathcal{B}} \sum_{i=1}^{\mathcal{B}} \bar{g}^t + \Delta g_i^t = \bar{g}^t + \frac{1}{\mathcal{B}} \sum_{i=1}^{\mathcal{B}} \Delta g_i^t. \tag{7}$$

Apply the $L^2$ Weak Law (Durrett, 2010, Theorem 2.2.3), we have: $g_{\mathcal{B}}^t \leq \bar{g}^t + \frac{\sigma^2}{\mathcal{B}}$, which can be also understood as:

$$\mathbb{E}_{x_i, y_i \in \{\mathcal{X}, \mathcal{Y}\}} \left[ \bar{g}^t - g_{\mathcal{B}}^t \right] \leq \sigma^2 / \mathcal{B} \tag{8}$$

## D   TIME VARYING COMPONENT BOUND

Consider the following equation of time-varying component:

$$\phi(t) = \frac{1}{\alpha + \exp\{\lambda(t - d_e)\}} + \gamma_{\min} \tag{9}$$

We then analyze its derivative (Equation (10))to gain deeper insights into how it can affect the gradient scaling ratio.

$$\frac{\partial \phi(t)}{\partial t} = \frac{-\lambda \exp\{\lambda t - \lambda d_e\}}{(\alpha + \exp\{\lambda t - \lambda d_e\})^2} \leq 0 \quad \text{w.r.t.} \quad \begin{cases} (\alpha + \exp\{\lambda t - \lambda d_e\})^2 \geq 0 \\ \lambda \exp\{\lambda t - \lambda d_e\} \geq 0 \end{cases} \tag{10}$$

Thus function $\phi(t)$ is a decreasing function for any $t \in [0, T)$. Therefore, the minimum value of the above function at $T \to \infty$ is $\gamma_{\min}$ as follows:

$$\min\{\phi(t)\} = \lim_{t \to \infty} \frac{1}{\alpha + \exp\{\lambda(t - d_e)\}} + \gamma_{\min} = \gamma_{\min} \tag{11}$$

While the maximum value at $t = 0$ is as follows:

$$\max\{\phi(t)\} = \phi(t = 0) = \frac{1}{\alpha + \exp\{-\lambda d_e\}} \tag{12}$$

Hence the time-varying component has the following bounds:

$$\gamma_{\min} \leq \phi(t) \leq \frac{1}{\alpha + \exp\{-\lambda d_e\}} \tag{13}$$

# E   DETAILED ANALYSIS

Overall, it is transparent that TVLARS performs well in most settings followed by LARS and LAMB, especially with batch sizes of 512, 1024, 2048, and 4096 (Table 1). In 512 sampled batch size experiments, TVLARS achieves 78.92, 81.42, and 81.56% on CIFAR10 and 37.56, 39.28, and 39.64% on Tiny ImageNet related to the LR of 1, 2, and 3, as opposed to 74.64, 77.49, and 79.64% and 33.72, 36.60, and 38.92% accomplished from LARS, respectively. TVLARS continues to perform better on even higher batch sizes: 1024, 2048, and 4096 with a stable rise in accuracy. While the performance of LARS varies approximately from 75% to 80%, TVLARS gains accuracy nearly or above 82% trained on CIFAR10 with batch sizes of 1024, 2048, and 4096. Furthermore, this assertion remains the same for experiments with Tiny ImageNet that TVLARS stably reaches the accuracy from 38% to 41%, in contrast with mostly from 33% to 38% of LARS. On even higher batch size settings: 8K and 16K, TVLARS achieves more stable performances: from approximately 81% to nearly 83%, and from 36% to above 42% on CIFAR10 and Tiny ImageNet, respectively. Although the performance of TVLARS and LARS are close to each other, the performance of the model trained by LARS accompanied by a warm-up strategy tends to explode and slow convergence in a very high batch size of 16K (see Figure 19c), which attains only 38% on CIFAR10 compared to fast convergence (within 20 epochs) and stable learning of TVLARS (see Figures 31c, 43c, 49c, 55c, 61c), which accomplishes 80.42%. LAMB, on the other hand, plunges to just above poor performance, especially just above 0% experimenting with the two lower LRs for a batch size of 16K. However, in the highest LR considered, LAMB achieves a performance of approximately 80%.

Another interesting point is that the higher the LR is, the faster the model reaches the optimal point, which claims that owing to the high LR at the initial training phase, the exploration of the loss landscape is enhanced by avoiding the model being trapped by the early poor sharp minima Dinh et al. (2017); Keskar et al. (2017). This phenomenon is indicated evidently in Figures of Section G, especially in experiments with a decay coefficient of 0.0001 used in Table 1. In detail, in the experiment with batch sizes of 512 and 1024, the convergence rate of the objective functions is nearly unchanged, which takes from 20 to 40 (see Figures 44, 45) epochs to reach the optimal point, compared to more than 40 in LARS with warm-up strategy (see Figures 14, 15, 16, 17, 18, 19). Moreover, the objective function value of TVLARS plunges to the optimal value after just above 20 epochs (47, 48, 49), as opposed to nearly 60 and 80 epochs in the experiments of LARS with batch sizes of 4K, 8K, and 16K (17a, 17b, 17c, 18a, 18b, 18c, 19a, 19b, 19c), respectively. This circumstance is caused by the imperfect exploration ability of a small LR Li et al. (2019); Keskar et al. (2017) in the warm-up phase, which steadily increases the LR from a small LR to a target LR by iteratively updating the LR $\gamma_t = \gamma_{\text{target}} \times \frac{t}{T}$. Although LARS and TVLARS ($B = 16K$) conducted with LRs of 15 and 17 reach the optimal point after 60 epochs, TVLARS experimented with a LR of 19 converges after 20 epochs, in contrast to 20 epochs from LARS. LAMB experiments are not an exception, they tend to reach the optimal point deliberately. Experimented with batch sizes of 4K, 8K, and 16K, for instance, LAMB acquires from 60 to 80 epochs to reach the optimum point for all LRs considered (11, 12, 13).

On the other hand, as the ability to avoid the many sharp minima that exist near the weight initialization point by high initial LR Hoffer et al. (2017), Kim et al. (2021), TVLARS enhances the model by finding a better performance local minima (see Figures 44a, 44b, 44c, 45a, 45b, 45c), which dives to 0.6 loss value, contrasted to scarcely higher than 1.0 achieved by LARS showed in Figure 14a, 14b, 14c, 15a, 15b, 15c. Especially in larger batch-size training ($B \in \{8K, 16K\}$), while TVLARS reach a loss value of under 0.75 (48a, 48b, 48c), LARS reaches to local minima whose performance of approximately 1.0 error (18a, 18b, 18c). Furthermore, TVLARS performance plunges to optimum (error $\approx 0$) (see Figures 49a, 49b, 49c), as opposed to significantly higher than 2 achieved by LARS (19a, 19b, 19c). As a result of the small LR, LARS with the warm-up strategy is effortlessly trapped by low-performance sharp minima. This problem is interpreted by the dramatically unpredictable oscillation of $\|\nabla_w \mathcal{L}(w)\|$ in the early phase, which is shown in Figures 2g, 2h, 2e. Since the norm of model layers' gradient $\|\nabla w\| = \{\|\nabla w_k\|\}_{k=0}^{K-1}$ varies from 0 to a much larger value frequently after several epochs, the model is stuck by local sharp minima whose different performance Keskar et al. (2017): larger than 20 error value. After 60 to 80 epochs, the model trained by LARS converges without being immobilized by poor sharp minima. This is a result of the ratio $\|w\|/\|\nabla w\| \to \infty$ as $\|\nabla w\| \to 0$, which creates a high adaptive LR to escape the trapped minima, where $\|w\|$ gradually decrease after each iteration. The higher the LR is, the more expeditious LARS achieves, which converges to a minimum whose loss value of 2 after 80, 70, and 60 epochs with LRs of 15, 17, and

19, separately. LAMB is not an exception, whose $\|w\|/\|\nabla w\|$ fluctuate frequently, which indicates the ability of LAMB to tackle the poor performance sharp minima, though its performance is not satisfactory that LAMB's loss value drops to around 1 (8, 9, 10, 11, 12, 13), compared to under 0.5 loss value achieved by TVLARS. However, in the experiments with the largest LRs considered, LAMB finds it easy to escape the poor sharp minima that the ratio $\|w\|/\|\nabla w\|$ is not oscillated drastically (see Figures 8c, 9c, 10c, 11c, 12c, 13c), and converge to optimum within 40 epochs whose performance are nearly the same with LARS.

## F  LARS WITH AND WITHOUT WARM-UP STRATEGY ADAPTIVE RATIO LNR

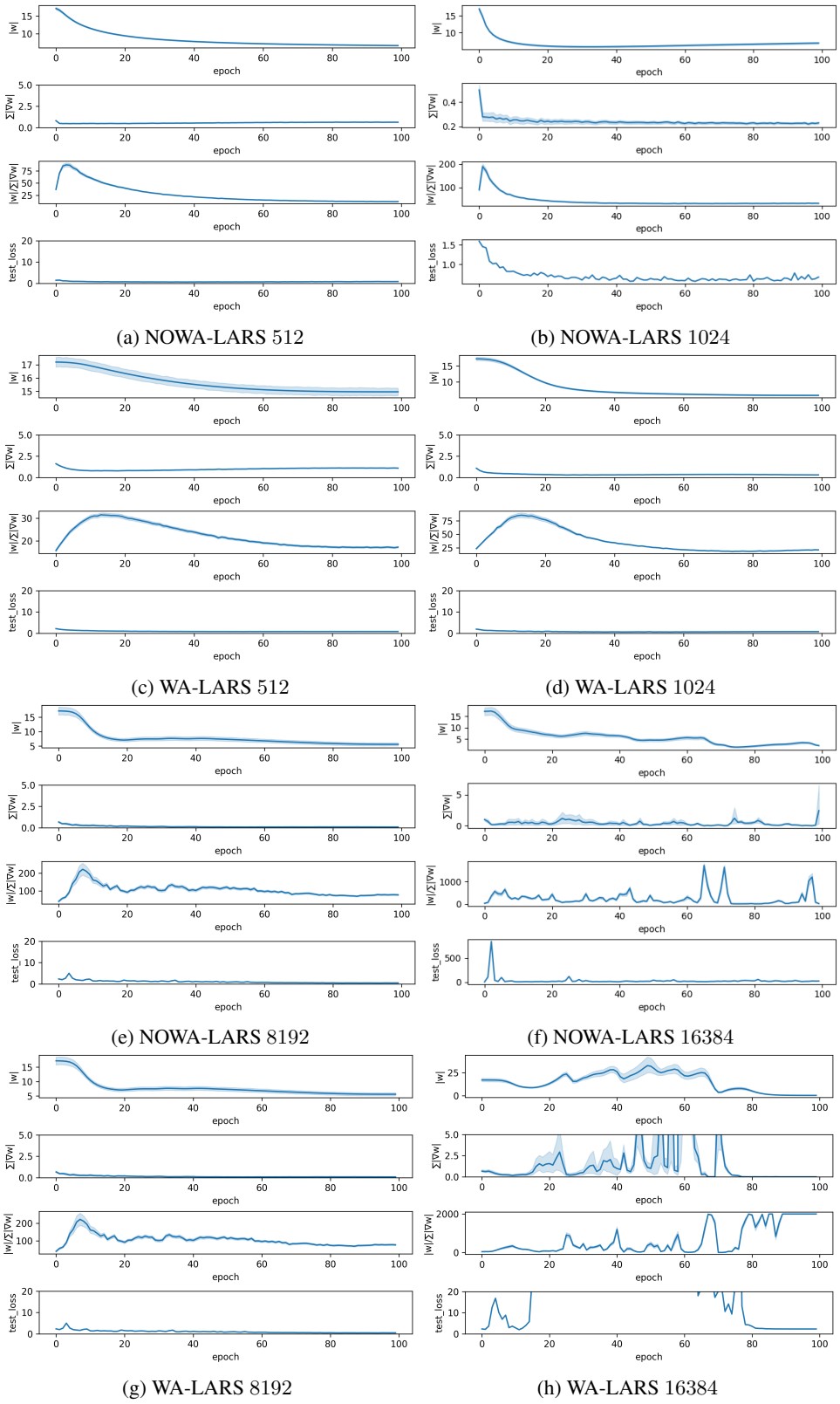

(a) NOWA-LARS 512

(b) NOWA-LARS 1024

(c) WA-LARS 512

(d) WA-LARS 1024

(e) NOWA-LARS 8192

(f) NOWA-LARS 16384

(g) WA-LARS 8192

(h) WA-LARS 16384

# G FURTHER EXPERIMENTS ON SCALING RATIO

## G.1 LAMB

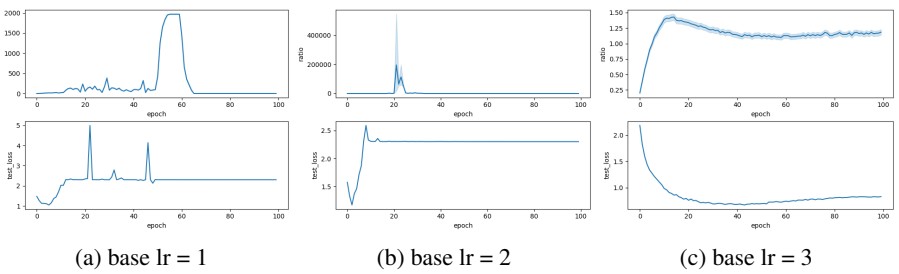

(a) base lr = 1          (b) base lr = 2          (c) base lr = 3

Figure 8: Batch size = 512

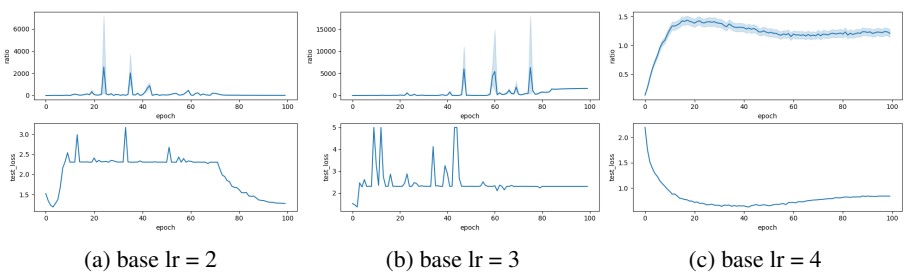

(a) base lr = 2          (b) base lr = 3          (c) base lr = 4

Figure 9: Batch size = 1024

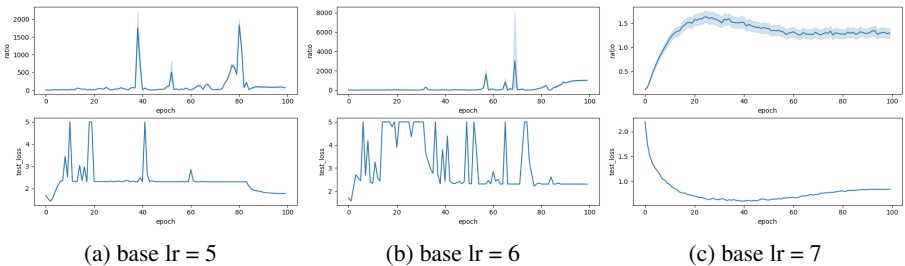

(a) base lr = 5          (b) base lr = 6          (c) base lr = 7

Figure 10: Batch size = 2048

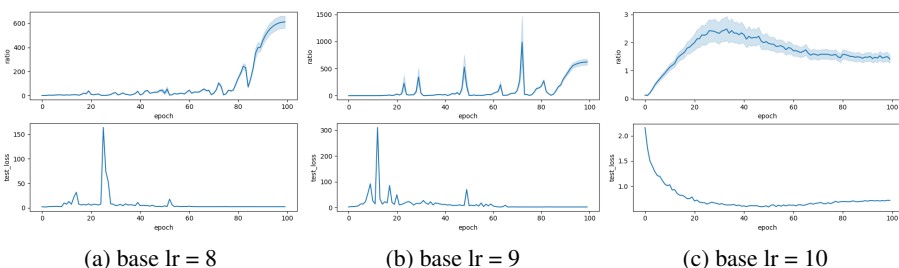

(a) base lr = 8          (b) base lr = 9          (c) base lr = 10

Figure 11: Batch size = 4096

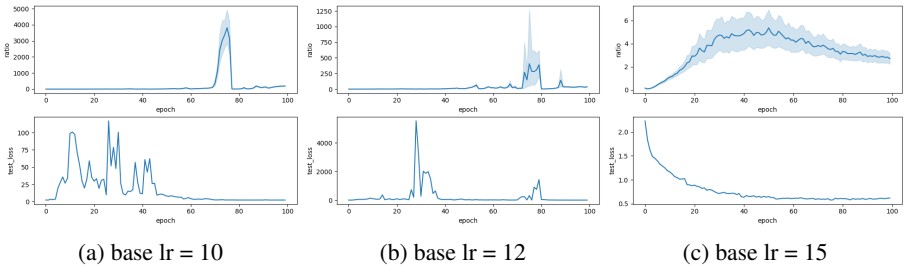

(a) base lr = 10        (b) base lr = 12        (c) base lr = 15

Figure 12: Batch size = 8K

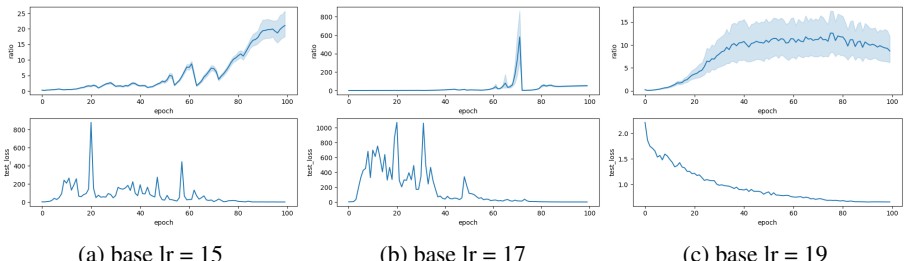

(a) base lr = 15        (b) base lr = 17        (c) base lr = 19

Figure 13: Batch size = 16K

## G.2 LARS

### G.2.1 WITH WARM UP

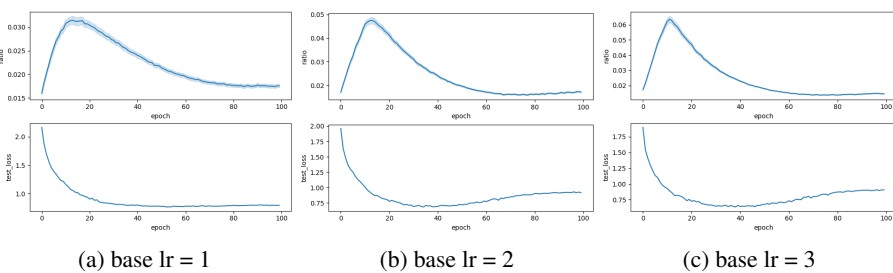

(a) base lr = 1      (b) base lr = 2      (c) base lr = 3

Figure 14: Batch size = 512

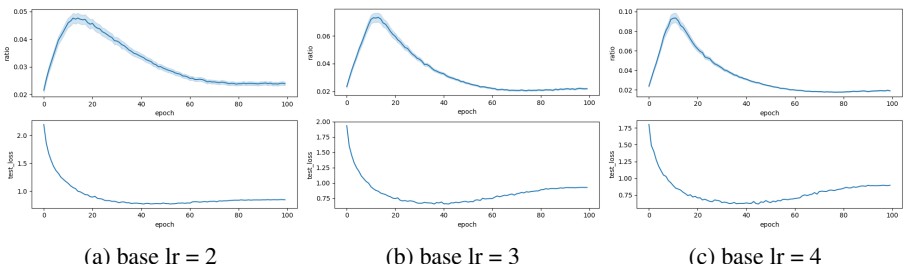

(a) base lr = 2      (b) base lr = 3      (c) base lr = 4

Figure 15: Batch size = 1024

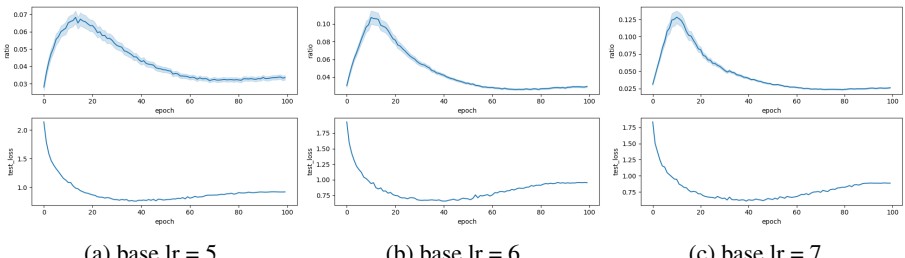

(a) base lr = 5      (b) base lr = 6      (c) base lr = 7

Figure 16: Batch size = 2048

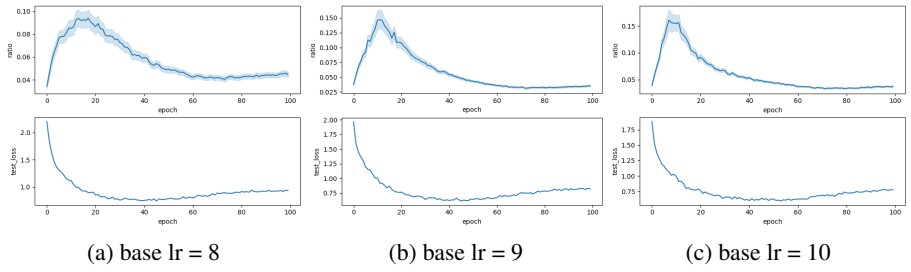

(a) base lr = 8    (b) base lr = 9    (c) base lr = 10

Figure 17: Batch size = 4096

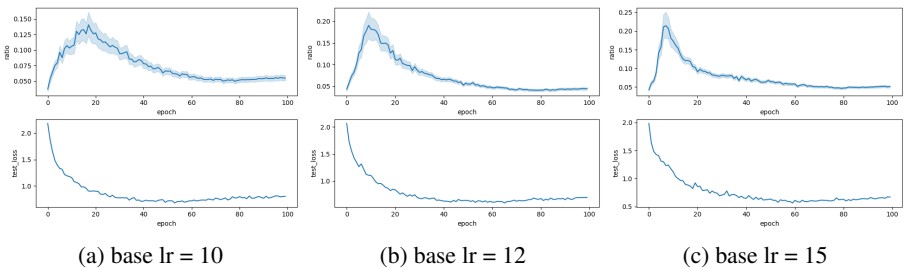

(a) base lr = 10    (b) base lr = 12    (c) base lr = 15

Figure 18: Batch size = 8192

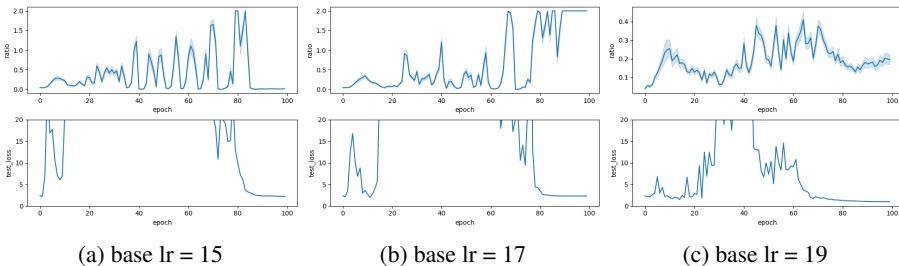

(a) base lr = 15    (b) base lr = 17    (c) base lr = 19

Figure 19: Batch size = 16384

### G.2.2 WITHOUT WARM UP

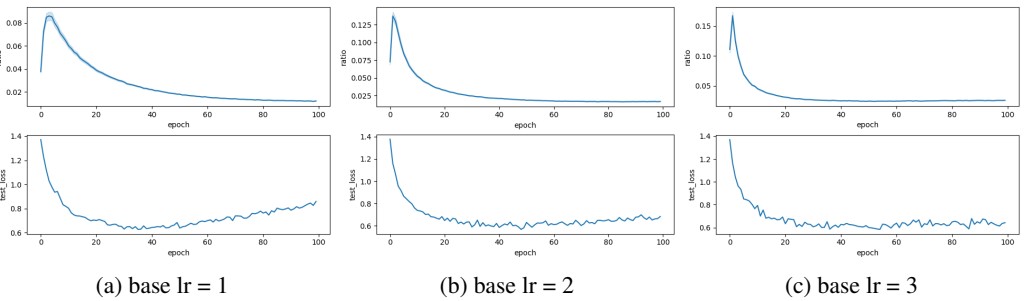

(a) base lr = 1    (b) base lr = 2    (c) base lr = 3

Figure 20: Batch size = 512

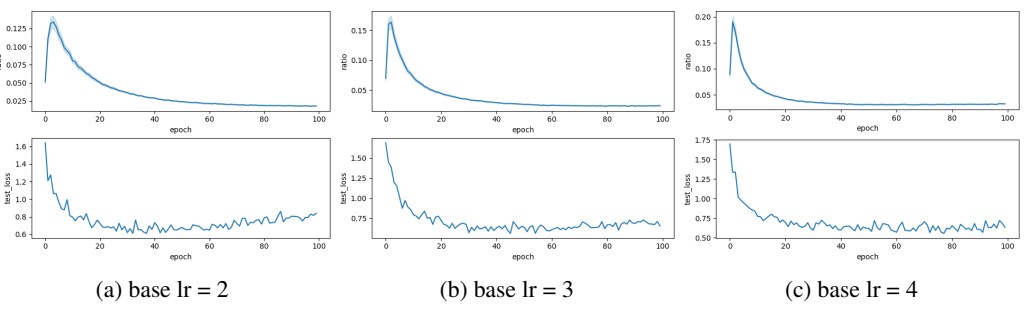

(a) base lr = 2    (b) base lr = 3    (c) base lr = 4

Figure 21: Batch size = 1024

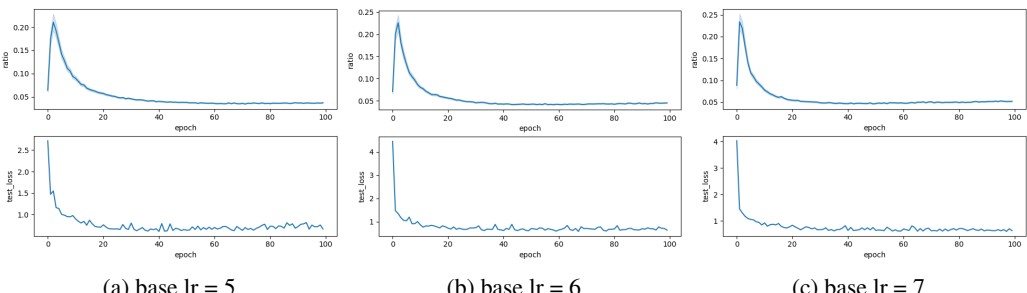

(a) base lr = 5    (b) base lr = 6    (c) base lr = 7

Figure 22: Batch size = 2048

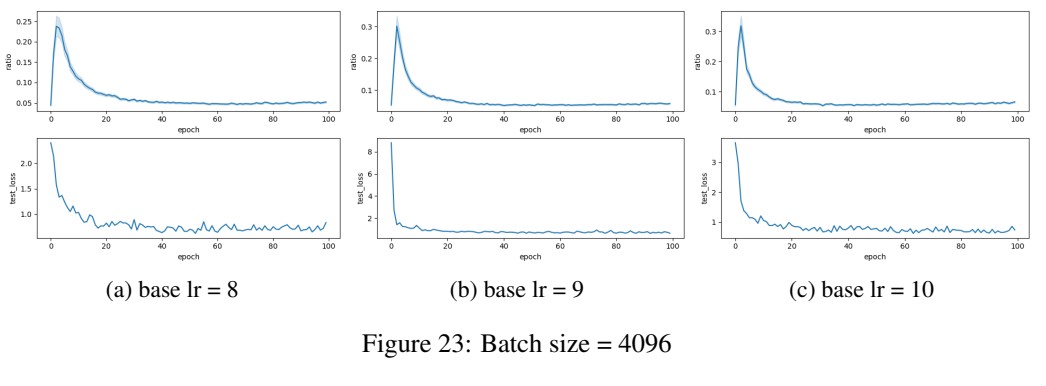

(a) base lr = 8            (b) base lr = 9            (c) base lr = 10

Figure 23: Batch size = 4096

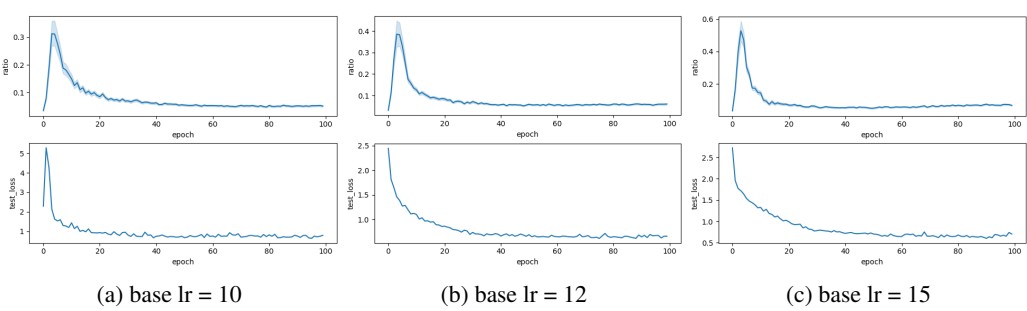

(a) base lr = 10           (b) base lr = 12           (c) base lr = 15

Figure 24: Batch size = 8192

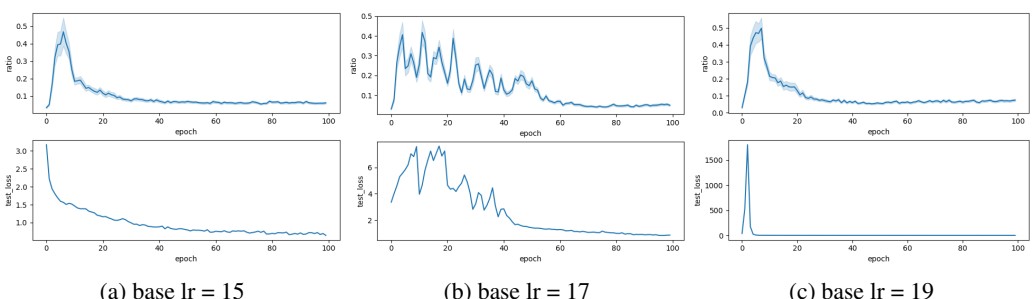

(a) base lr = 15           (b) base lr = 17           (c) base lr = 19

Figure 25: Batch size = 16384

### G.3 TVLARS

### G.3.1 $\lambda = 0.01$

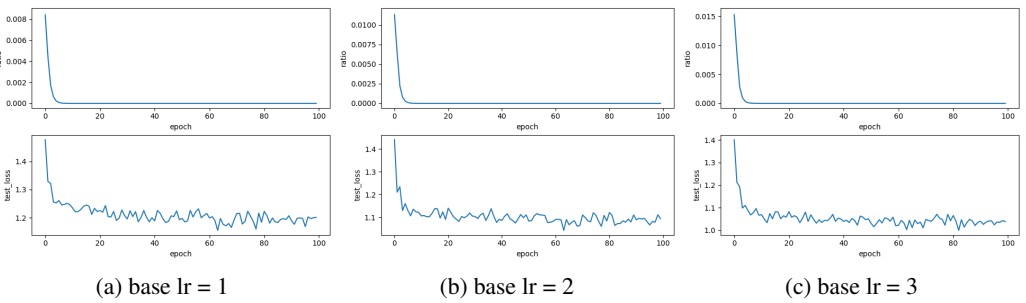

(a) base lr = 1

(b) base lr = 2

(c) base lr = 3

Figure 26: Batch size = 512

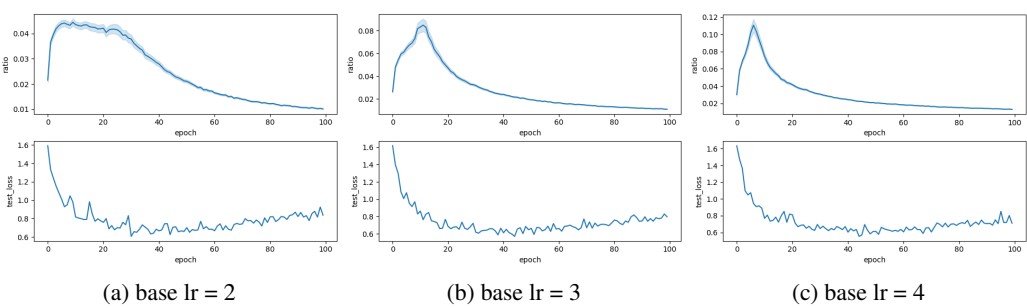

(a) base lr = 2

(b) base lr = 3

(c) base lr = 4

Figure 27: Batch size = 1024

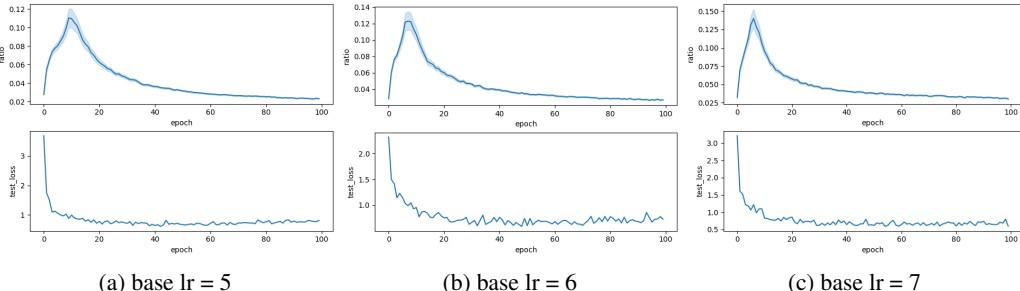

(a) base lr = 5

(b) base lr = 6

(c) base lr = 7

Figure 28: Batch size = 2048

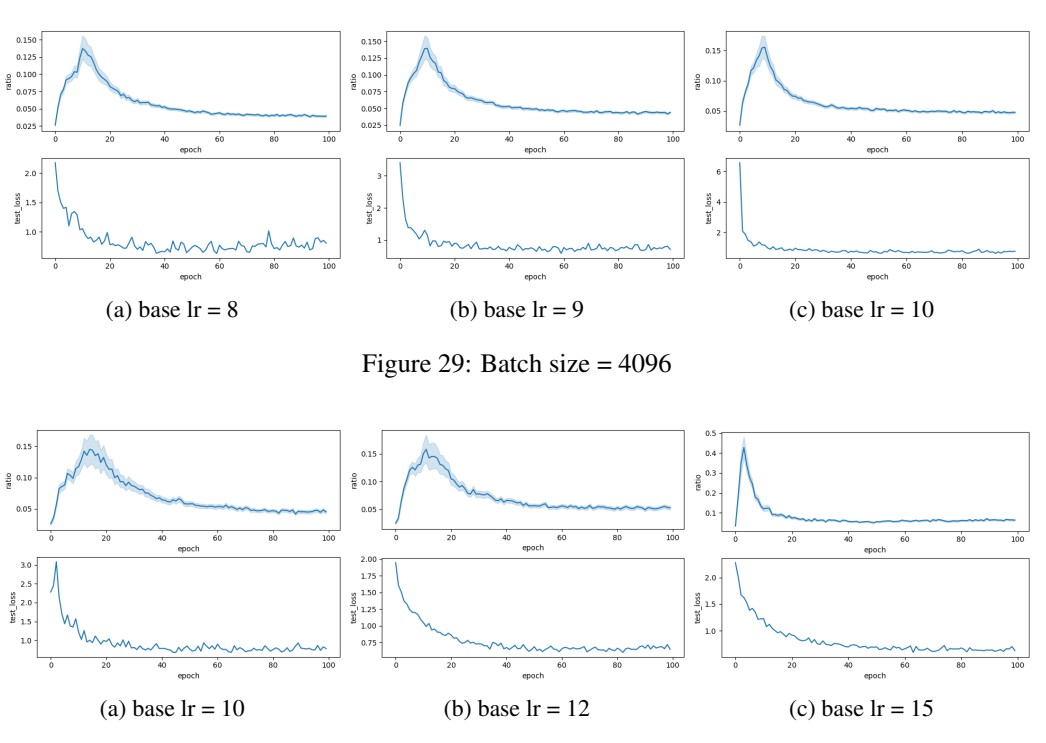

(a) base lr = 8      (b) base lr = 9      (c) base lr = 10

Figure 29: Batch size = 4096

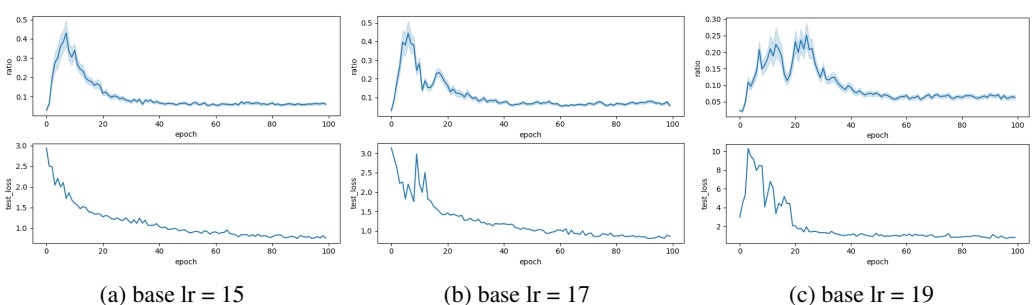

(a) base lr = 10      (b) base lr = 12      (c) base lr = 15

Figure 30: Batch size = 8192

(a) base lr = 15      (b) base lr = 17      (c) base lr = 19

Figure 31: Batch size = 16384

### G.3.2 $\lambda = 0.005$

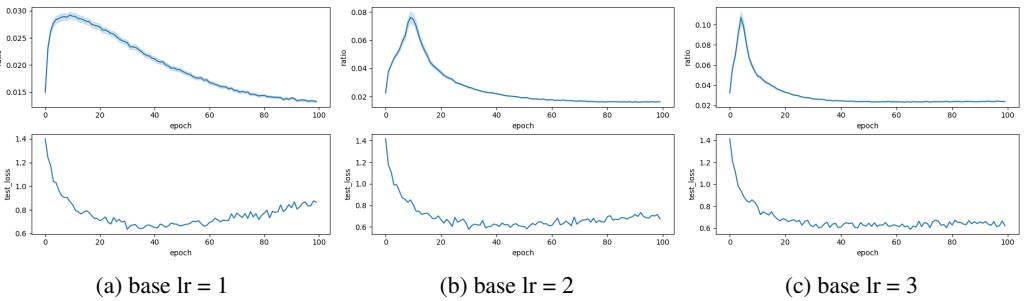

(a) base lr = 1

(b) base lr = 2

(c) base lr = 3

Figure 32: Batch size = 512

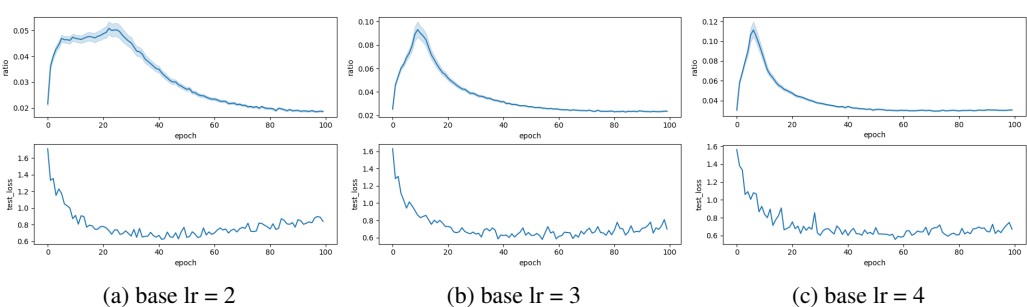

(a) base lr = 2

(b) base lr = 3

(c) base lr = 4

Figure 33: Batch size = 1024

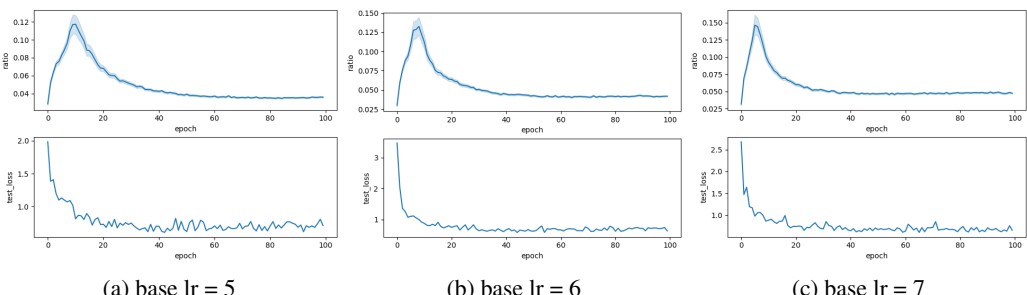

(a) base lr = 5

(b) base lr = 6

(c) base lr = 7

Figure 34: Batch size = 2048

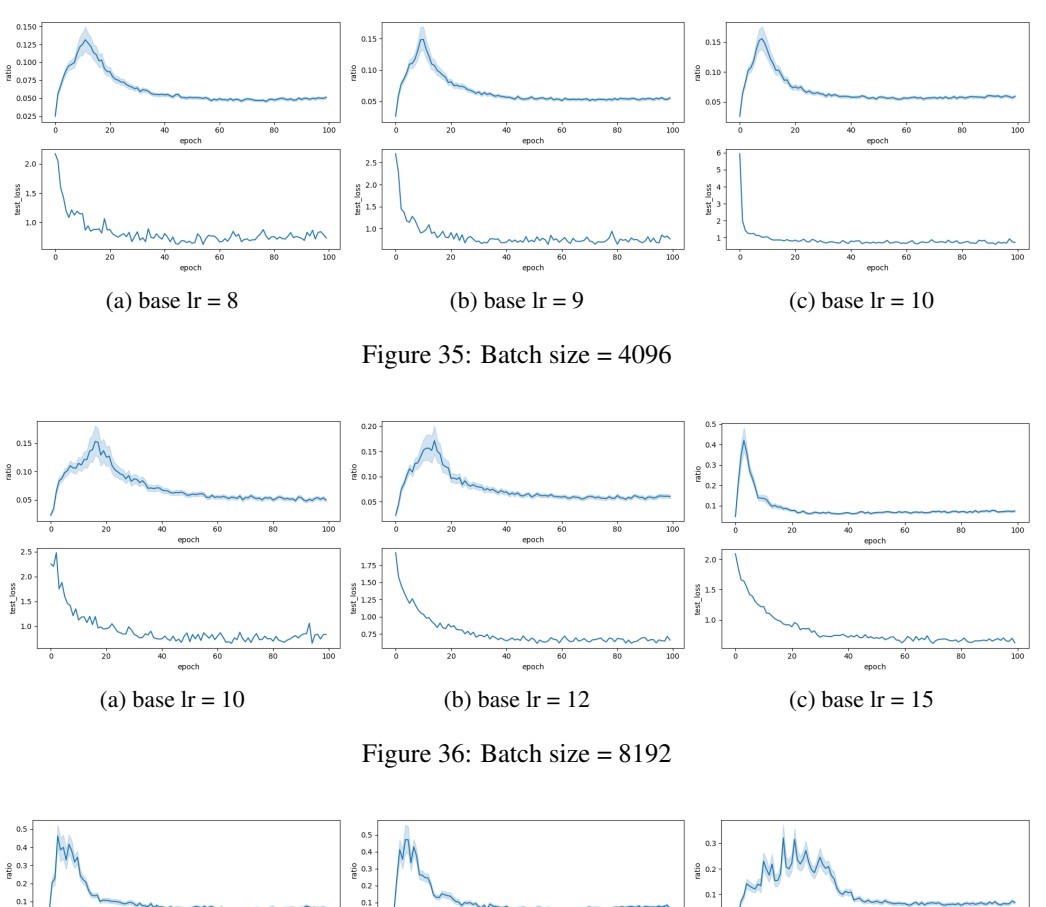

(a) base lr = 8      (b) base lr = 9      (c) base lr = 10

Figure 35: Batch size = 4096

(a) base lr = 10      (b) base lr = 12      (c) base lr = 15

Figure 36: Batch size = 8192

(a) base lr = 15      (b) base lr = 17      (c) base lr = 19

Figure 37: Batch size = 16384

### G.3.3 $\lambda = 0.001$

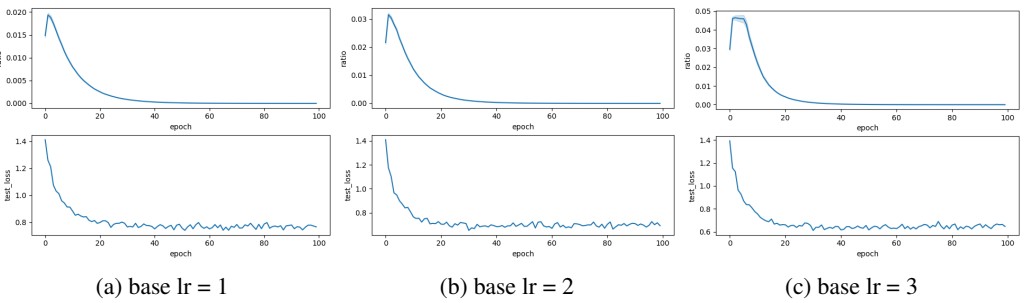

(a) base lr = 1      (b) base lr = 2      (c) base lr = 3

Figure 38: Batch size = 512

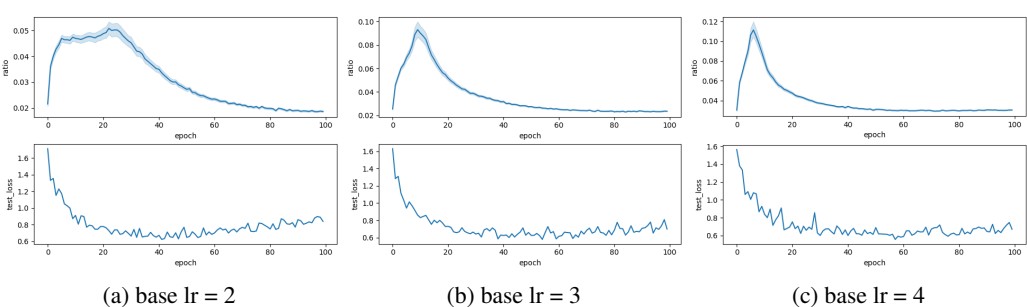

(a) base lr = 2      (b) base lr = 3      (c) base lr = 4

Figure 39: Batch size = 1024

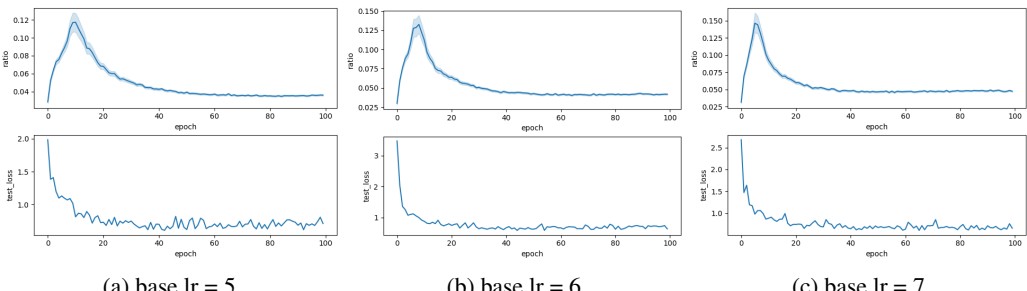

(a) base lr = 5      (b) base lr = 6      (c) base lr = 7

Figure 40: Batch size = 2048

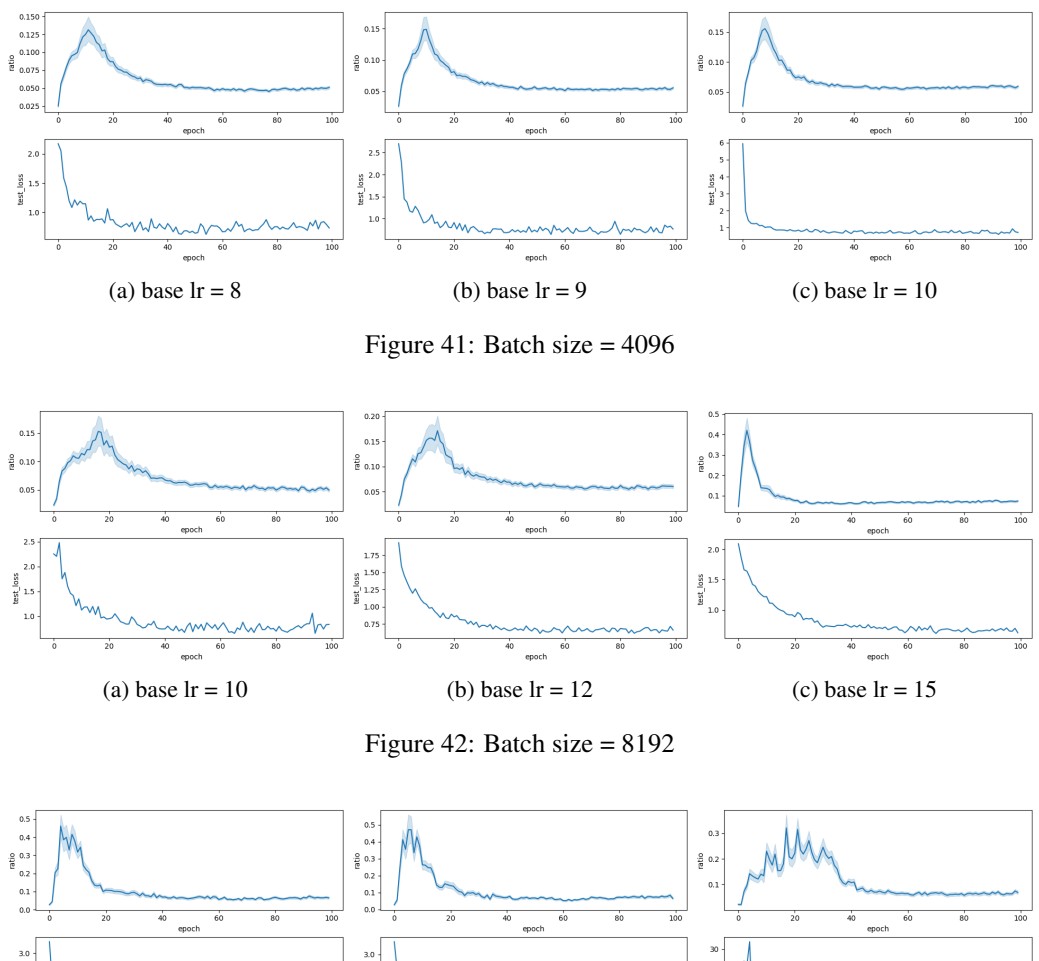

(a) base lr = 8      (b) base lr = 9      (c) base lr = 10

Figure 41: Batch size = 4096

(a) base lr = 10      (b) base lr = 12      (c) base lr = 15

Figure 42: Batch size = 8192

(a) base lr = 15      (b) base lr = 17      (c) base lr = 19

Figure 43: Batch size = 16384

### G.3.4 $\lambda = 0.0001$

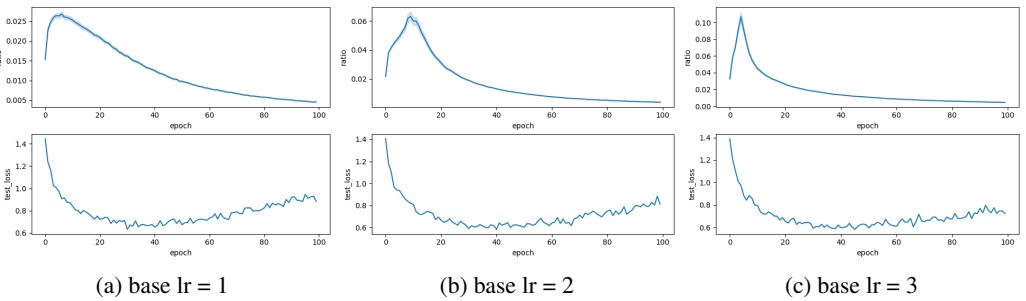

(a) base lr = 1        (b) base lr = 2        (c) base lr = 3

Figure 44: Batch size = 512

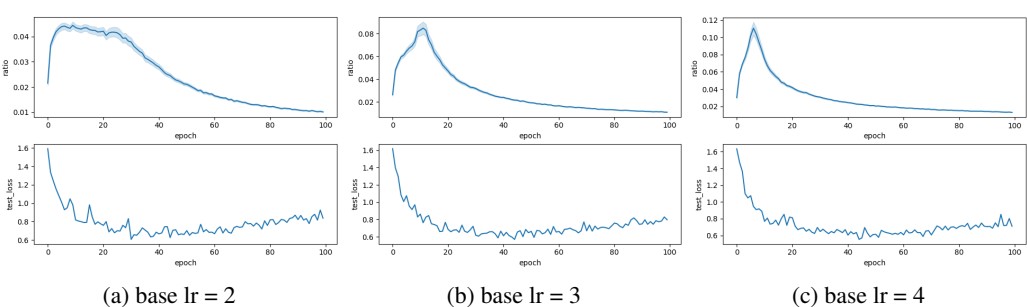

(a) base lr = 2        (b) base lr = 3        (c) base lr = 4

Figure 45: Batch size = 1024

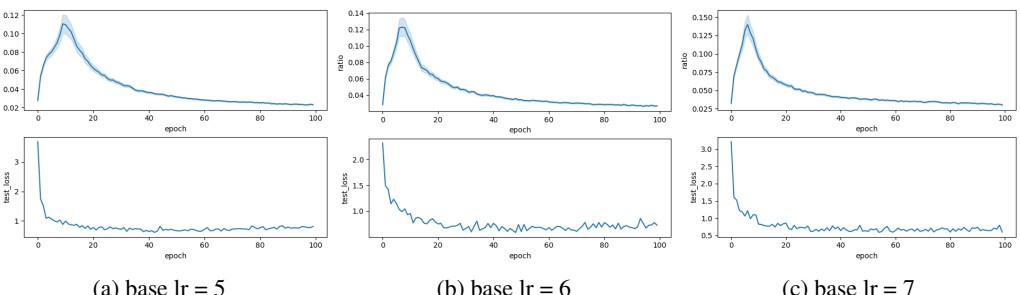

(a) base lr = 5        (b) base lr = 6        (c) base lr = 7

Figure 46: Batch size = 2048

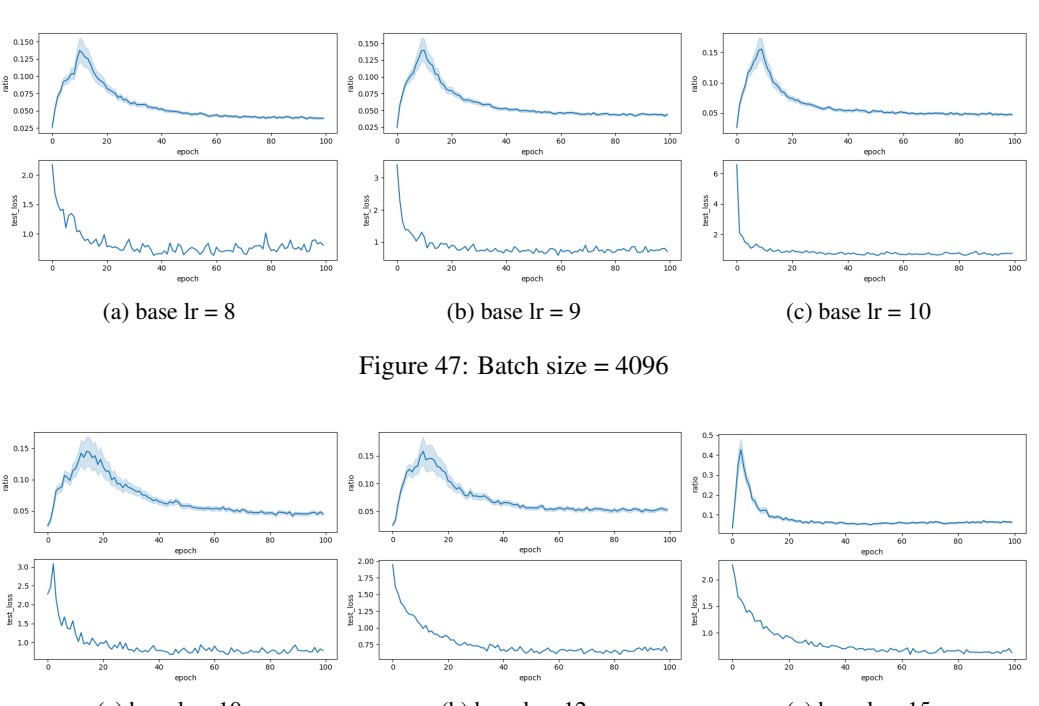

(a) base lr = 8         (b) base lr = 9         (c) base lr = 10

Figure 47: Batch size = 4096

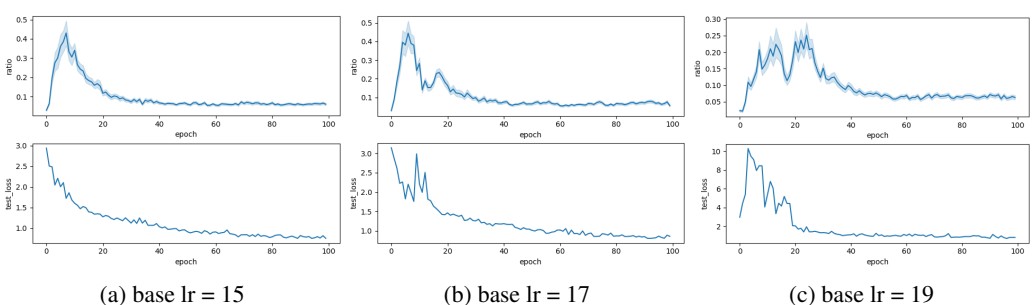

(a) base lr = 10        (b) base lr = 12        (c) base lr = 15

Figure 48: Batch size = 8192

(a) base lr = 15        (b) base lr = 17        (c) base lr = 19

Figure 49: Batch size = 16384

### G.3.5  $\lambda = 0.00001$

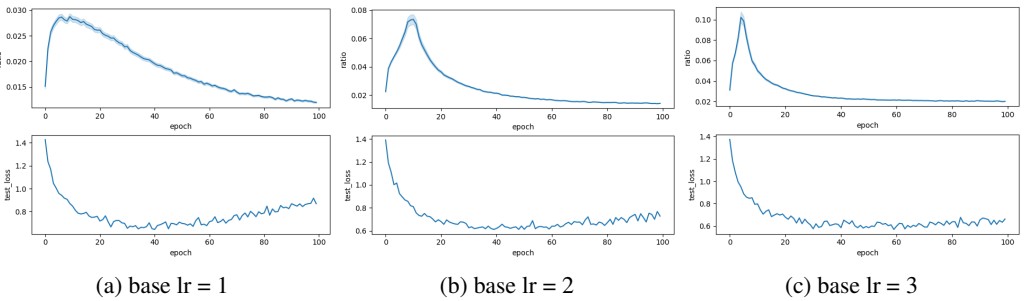

(a) base lr = 1

(b) base lr = 2

(c) base lr = 3

Figure 50: Batch size = 512

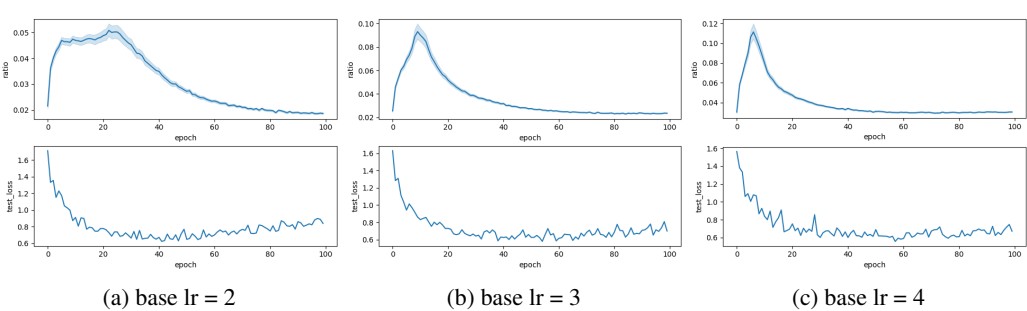

(a) base lr = 2

(b) base lr = 3

(c) base lr = 4

Figure 51: Batch size = 1024

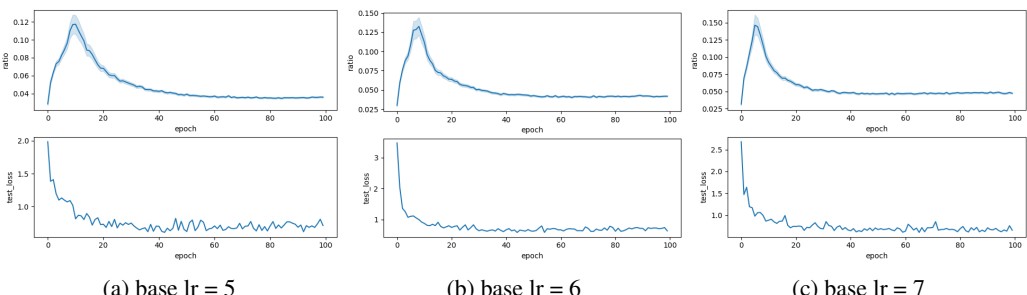

(a) base lr = 5

(b) base lr = 6

(c) base lr = 7

Figure 52: Batch size = 2048

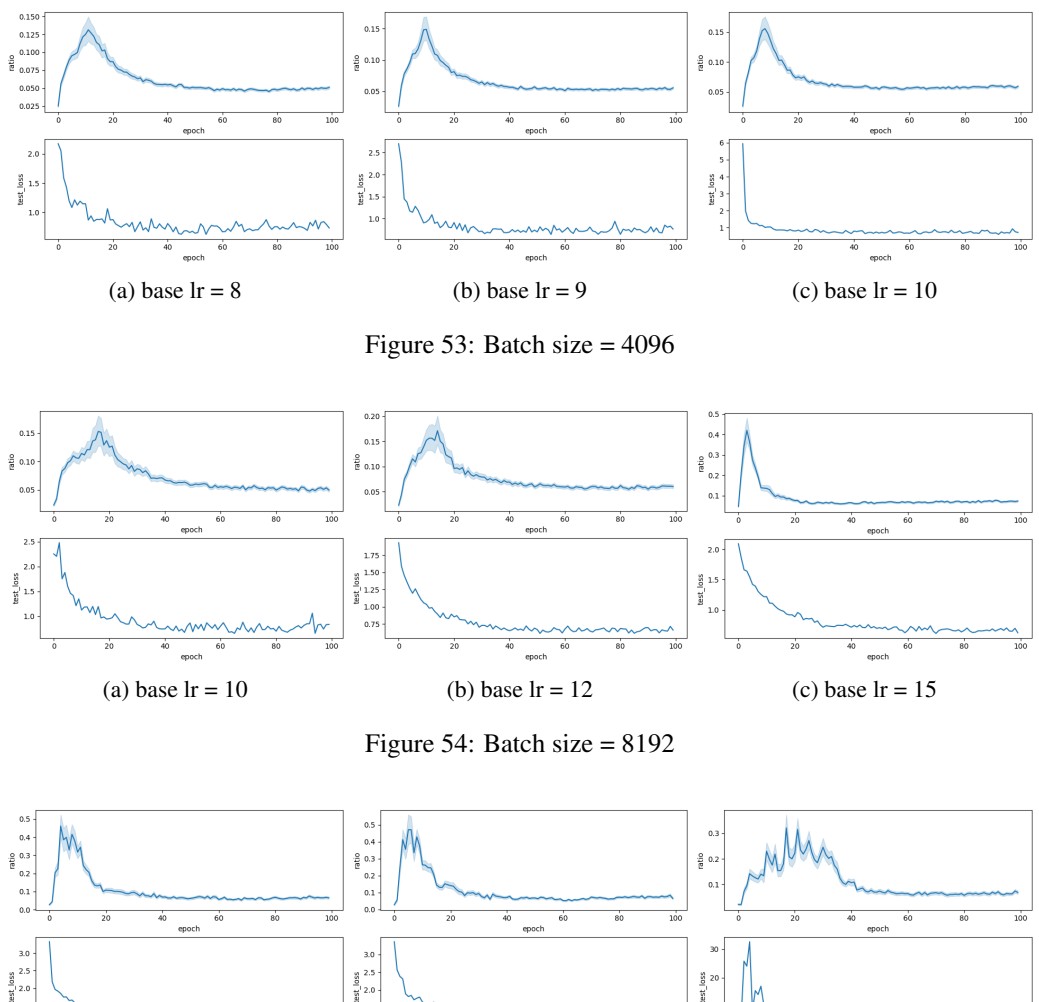

(a) base lr = 8      (b) base lr = 9      (c) base lr = 10

Figure 53: Batch size = 4096

(a) base lr = 10      (b) base lr = 12      (c) base lr = 15

Figure 54: Batch size = 8192

(a) base lr = 15      (b) base lr = 17      (c) base lr = 19

Figure 55: Batch size = 16384

### G.3.6 $\lambda = 0.000001$

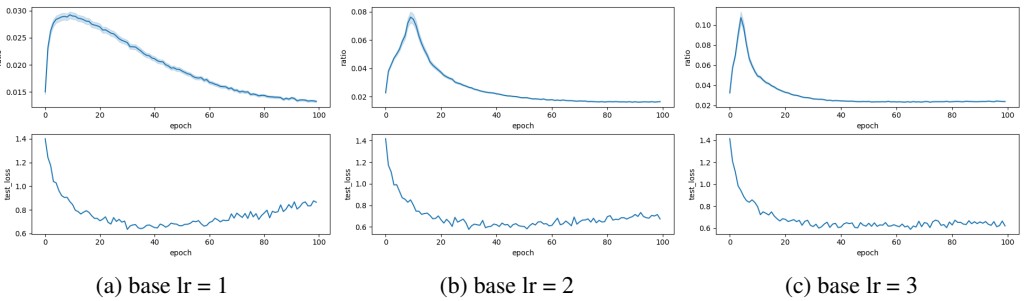

(a) base lr = 1

(b) base lr = 2

(c) base lr = 3

Figure 56: Batch size = 512

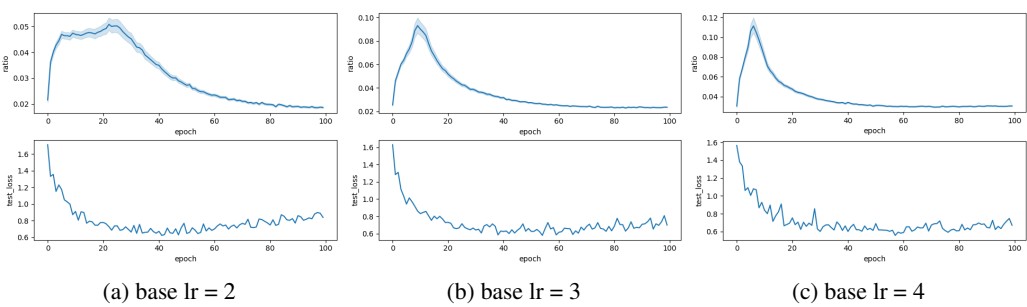

(a) base lr = 2

(b) base lr = 3

(c) base lr = 4

Figure 57: Batch size = 1024

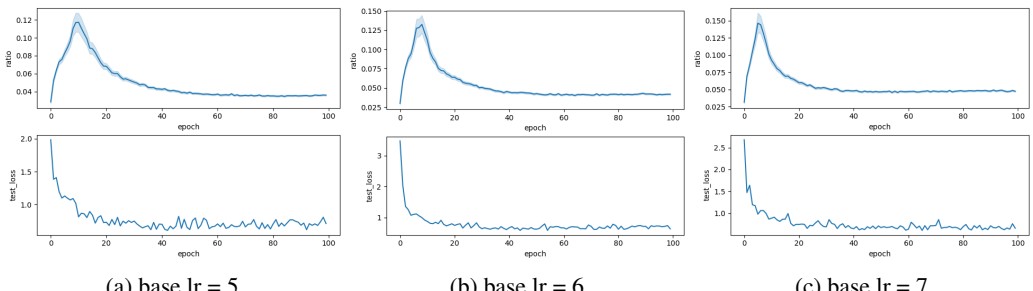

(a) base lr = 5

(b) base lr = 6

(c) base lr = 7

Figure 58: Batch size = 2048

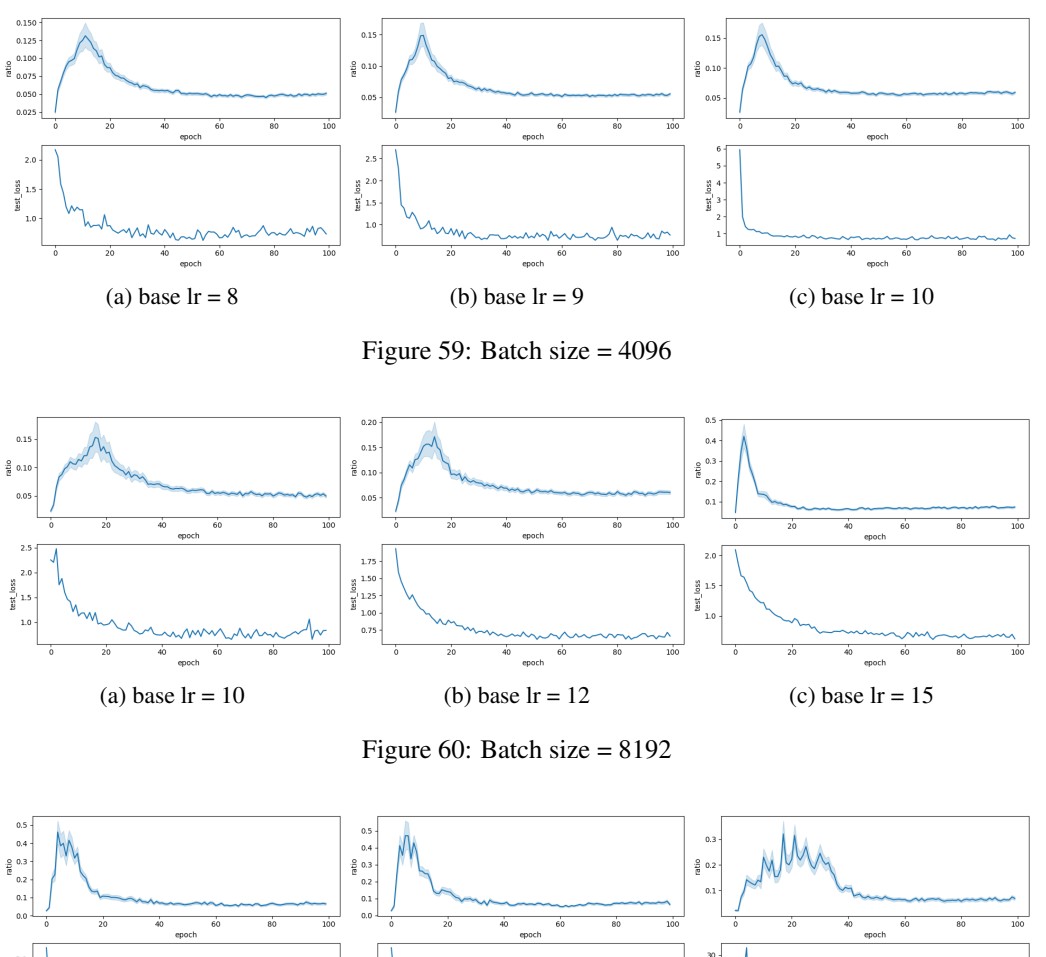

(a) base lr = 8        (b) base lr = 9        (c) base lr = 10

Figure 59: Batch size = 4096

(a) base lr = 10        (b) base lr = 12        (c) base lr = 15

Figure 60: Batch size = 8192

(a) base lr = 15        (b) base lr = 17        (c) base lr = 19

Figure 61: Batch size = 16384

# H FURTHER EXPERIMENT ON LARS WITH AND WITHOUT WARM-UP STRATEGY

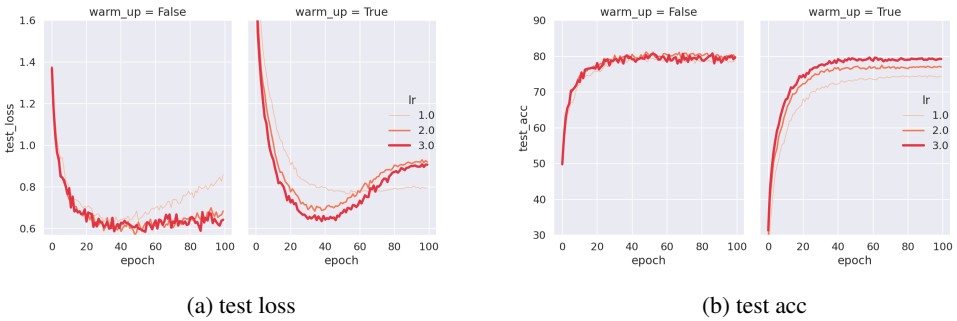

(a) test loss

(b) test acc

Figure 62: $\mathcal{B} = 512$

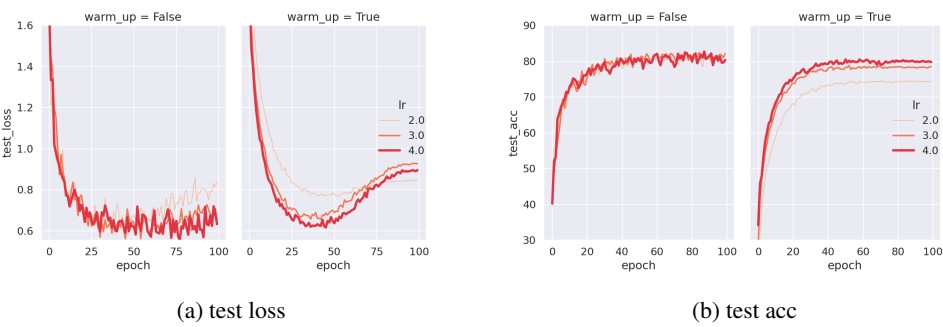

(a) test loss

(b) test acc

Figure 63: $\mathcal{B} = 1024$

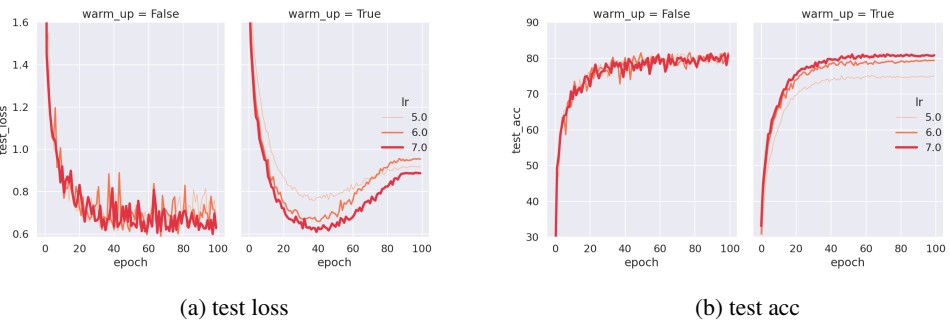

(a) test loss

(b) test acc

Figure 64: $\mathcal{B} = 2048$

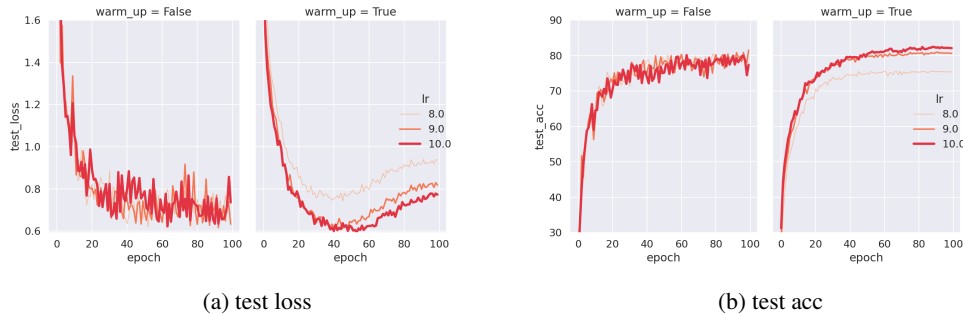

(a) test loss

(b) test acc

Figure 65: $\mathcal{B} = 4096$

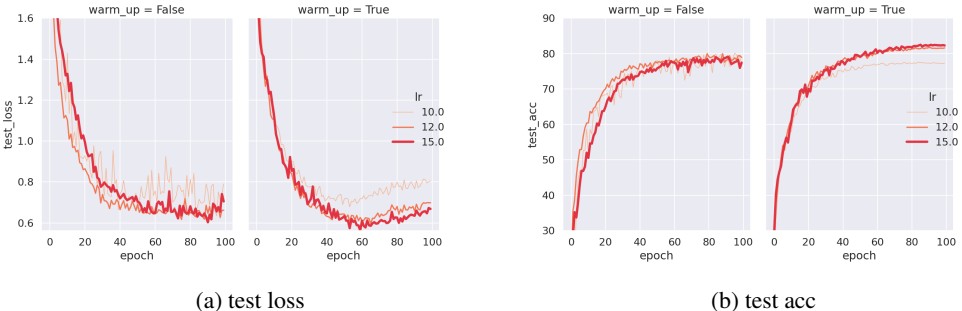

(a) test loss

(b) test acc

Figure 66: $\mathcal{B} = 8192$

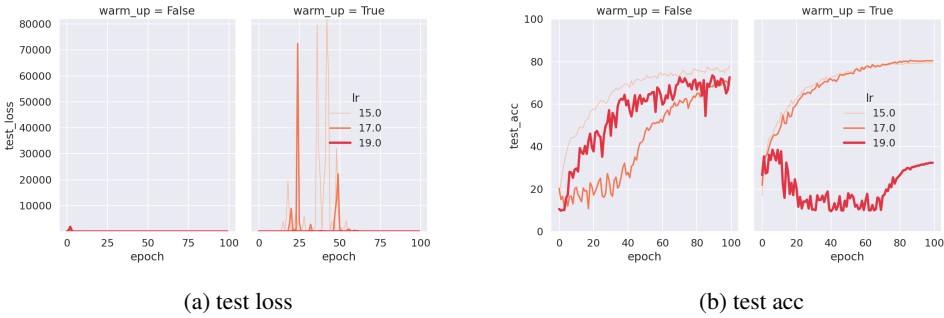

(a) test loss

(b) test acc

Figure 67: $\mathcal{B} = 16384$

# I   FURTHER EXPERIMENT OF TVLARS WITH A LARGE VARIATION OF DECAY COEFFICIENT

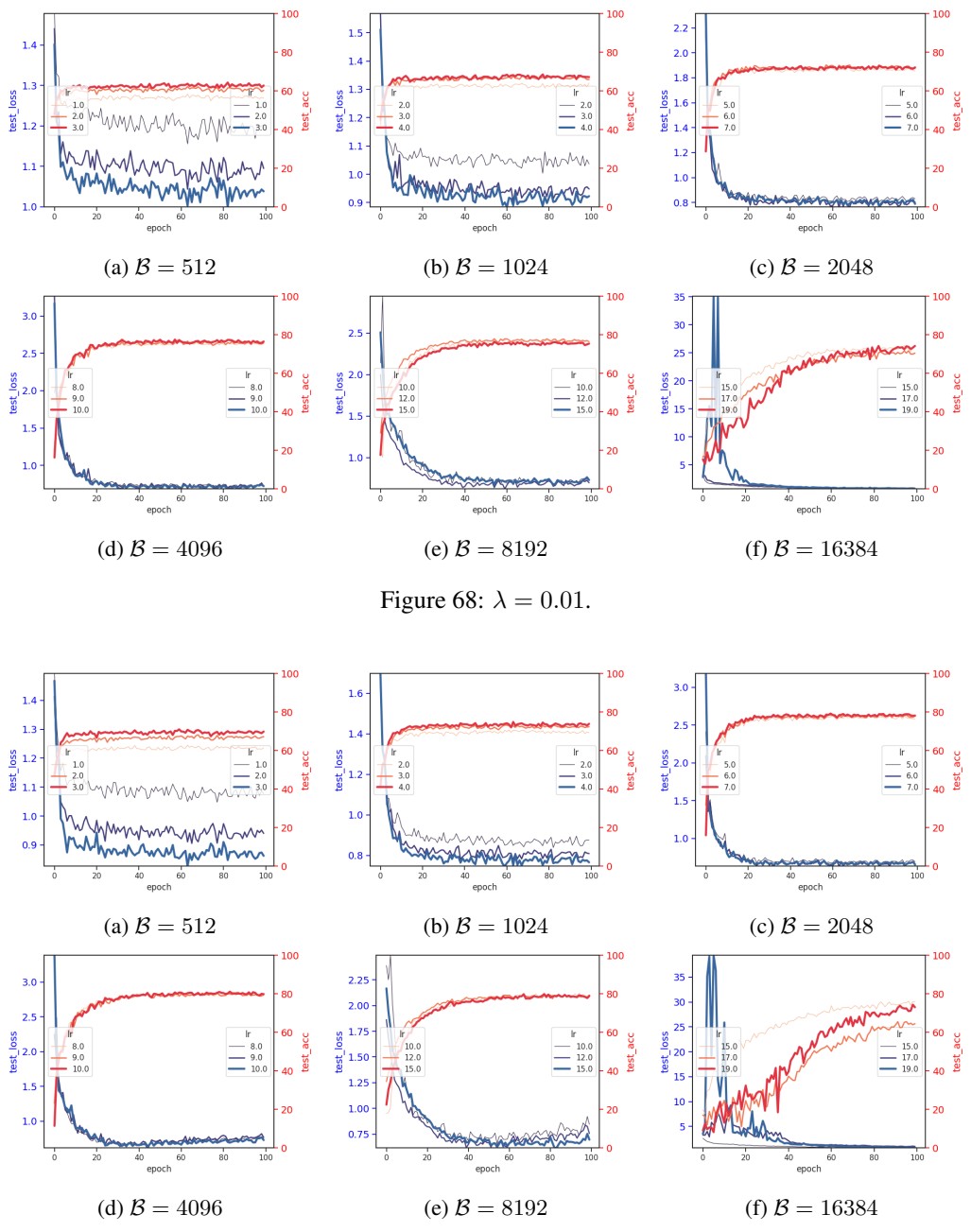

(a) $\mathcal{B} = 512$      (b) $\mathcal{B} = 1024$      (c) $\mathcal{B} = 2048$

(d) $\mathcal{B} = 4096$      (e) $\mathcal{B} = 8192$      (f) $\mathcal{B} = 16384$

Figure 68: $\lambda = 0.01$.

(a) $\mathcal{B} = 512$      (b) $\mathcal{B} = 1024$      (c) $\mathcal{B} = 2048$

(d) $\mathcal{B} = 4096$      (e) $\mathcal{B} = 8192$      (f) $\mathcal{B} = 16384$

Figure 69: $\lambda = 0.01$.

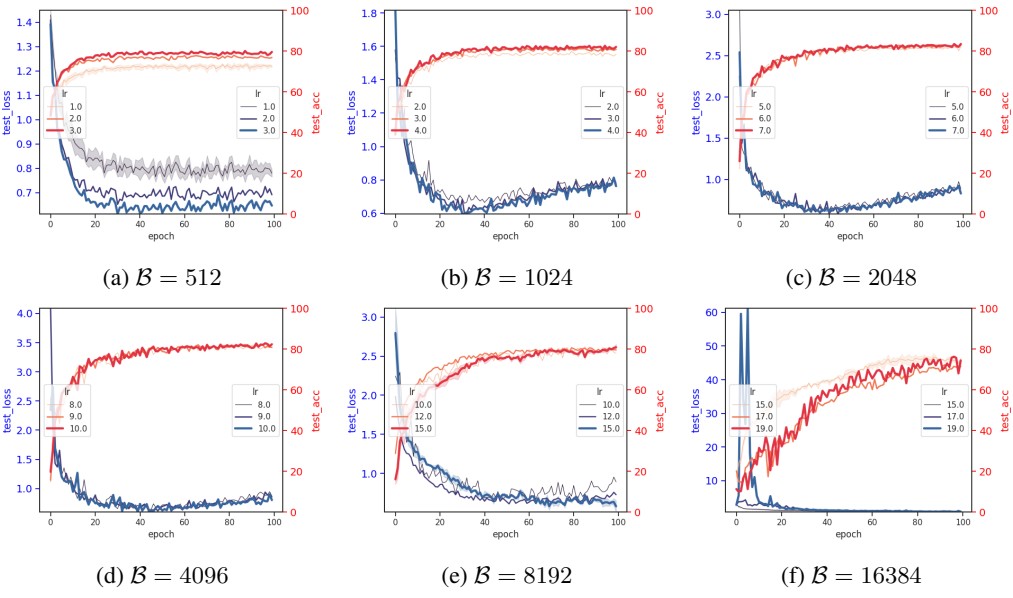

(a) $\mathcal{B} = 512$     (b) $\mathcal{B} = 1024$     (c) $\mathcal{B} = 2048$

(d) $\mathcal{B} = 4096$     (e) $\mathcal{B} = 8192$     (f) $\mathcal{B} = 16384$

Figure 70: $\lambda = 0.001$.

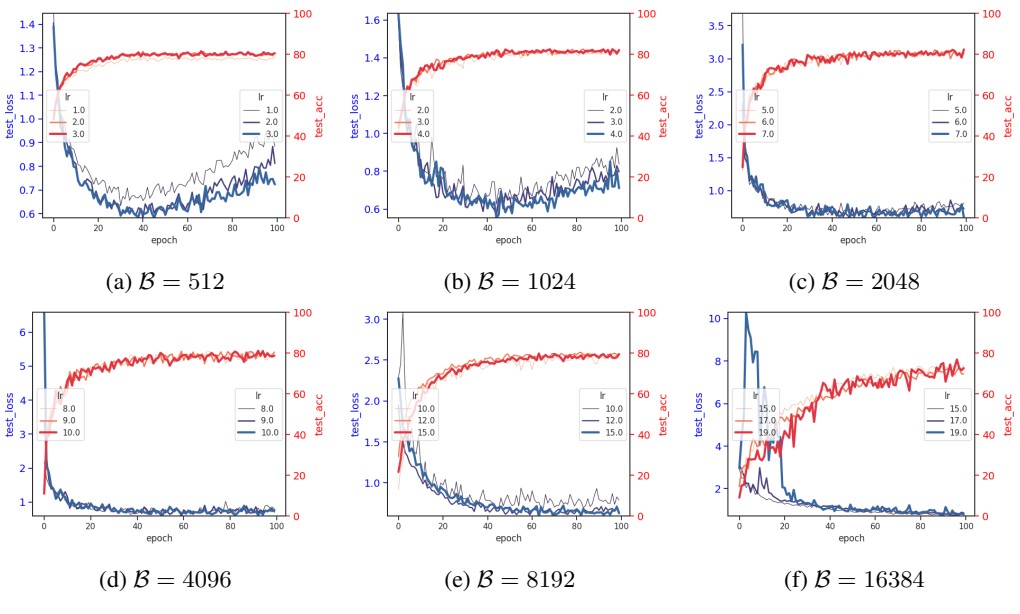

(a) $\mathcal{B} = 512$     (b) $\mathcal{B} = 1024$     (c) $\mathcal{B} = 2048$

(d) $\mathcal{B} = 4096$     (e) $\mathcal{B} = 8192$     (f) $\mathcal{B} = 16384$

Figure 71: $\lambda = 0.0001$.

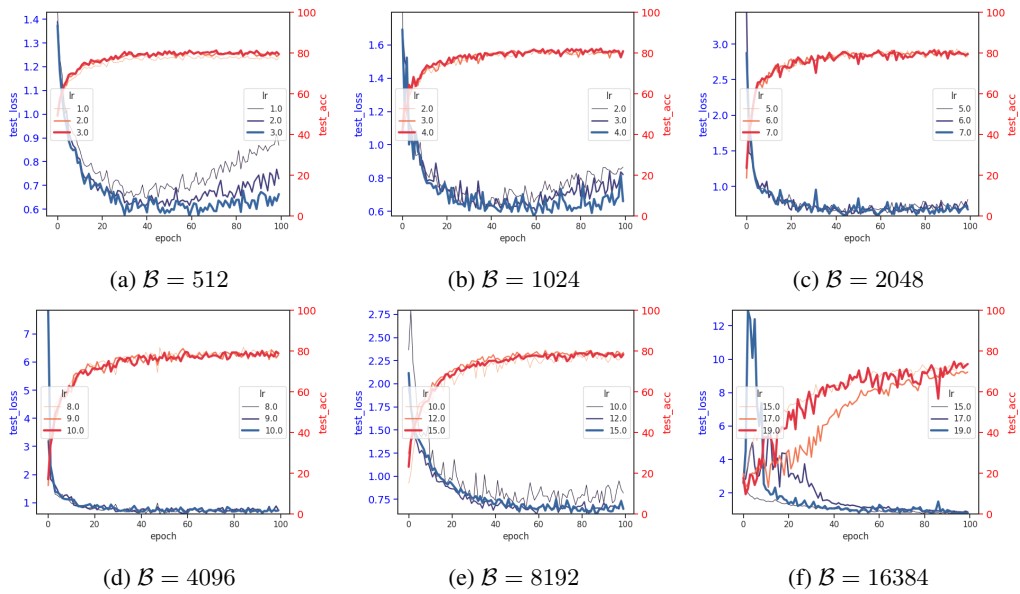

(a) $\mathcal{B} = 512$     (b) $\mathcal{B} = 1024$     (c) $\mathcal{B} = 2048$

(d) $\mathcal{B} = 4096$     (e) $\mathcal{B} = 8192$     (f) $\mathcal{B} = 16384$

Figure 72: $\lambda = 0.00001$.

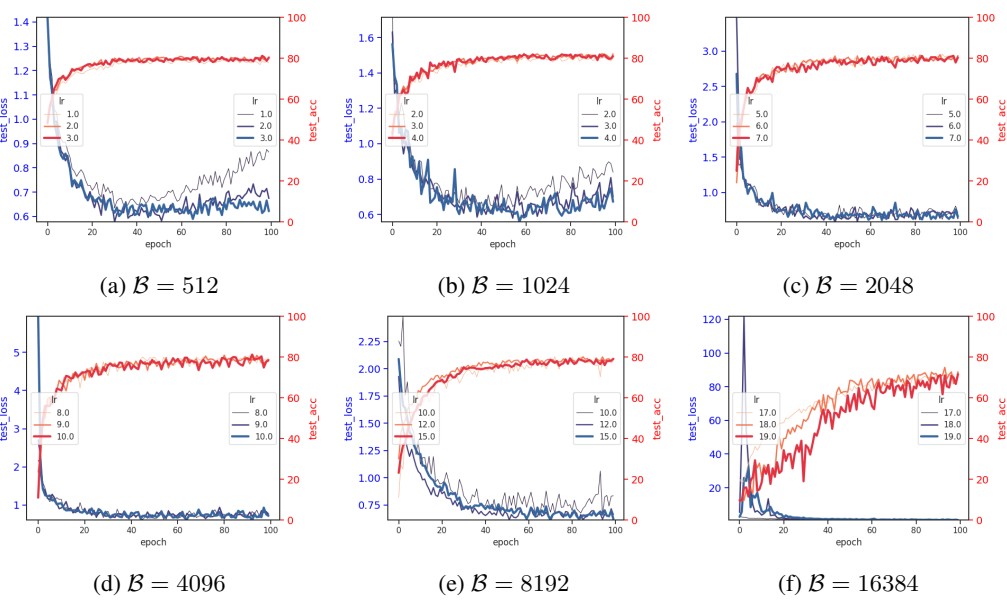

(a) $\mathcal{B} = 512$     (b) $\mathcal{B} = 1024$     (c) $\mathcal{B} = 2048$

(d) $\mathcal{B} = 4096$     (e) $\mathcal{B} = 8192$     (f) $\mathcal{B} = 16384$

Figure 73: $\lambda = 0.000001$.

# J  LEARNING RATE ANNEALED BY WARMUP STRATEGY

The figure 74 reveals the LLR in WA-LARS. As we can see from the figure, the LLR tends to be excessively low in the first 20 learning epochs in many DNN layers (i.e., layer 1 to layer 12). In the last layer, the LLR is nearly 0 in first 40 epochs. This problem leads to the gradient dismishing when applying the LLR to the back-propagation, which can be considered as redundant learning phase in WA-LARS.

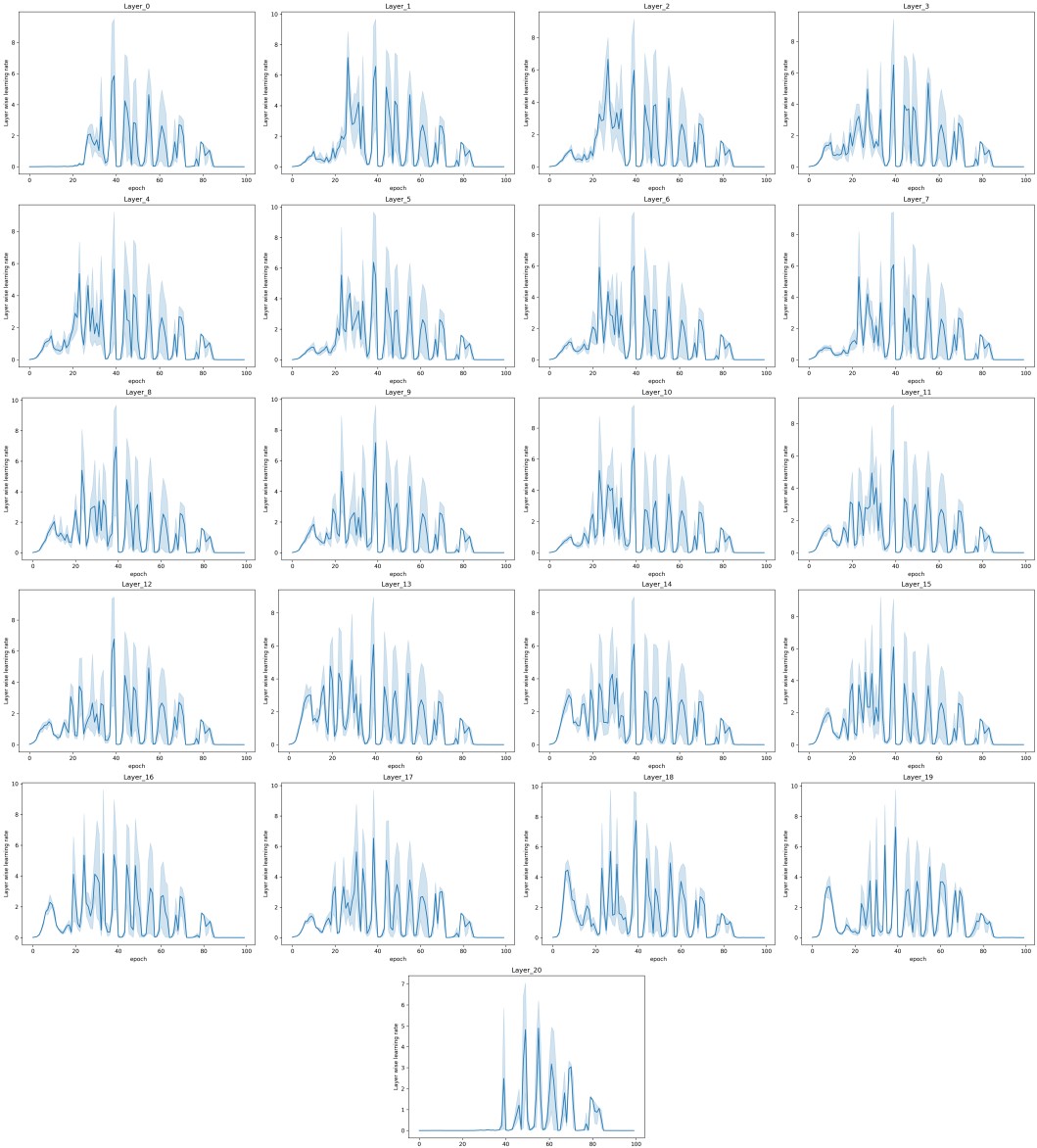

Figure 74: Learning rate annealed by warm-up strategy with a setting of 16K batches in every 20 layers of the model.

