# OpenReview forum: "REVISITING LARS FOR LARGE BATCH TRAINING GENERALIZATION OF NEURAL NETWORKS"
_ICLR.cc/2024/Conference — Submitted to ICLR 2024_

### Official Review · Reviewer_k26Y · 2023-10-27

**Soundness:** 4 excellent
**Presentation:** 4 excellent
**Contribution:** 2 fair
**Rating:** 6
**Confidence:** 4

**Summary:**

LARS proportionally scale the learning rate at each layer to improve stability in large batch learning. However, it is well known that LARS suffer from sluggish convergence at early stages of training, hence requiring a warmup procedure. This paper hypothesizes and empirically verifies that LARS tend to fall into sharp minimizers at the initial stages of training, leading to an explosion of the adaptively scaled gradient. As a result, this paper proposes TVLARS, and verify on experiments that TVLARS outperform LARS with warmup.

**Strengths:**

This paper proposes TVLARS, and demonstrates empirically that TVLARS outperform LARS under large batch settings for the CIFAR10 and TinyImageNet datasets. This paper is well-written, easy to follow, and the experiment results are clear.

**Weaknesses:**

The main weakness of this paper is that I am unsure whether TVLARS could be applied in big datasets that might benefit more from large batch training. After all, CIFAR and TinyImageNet are becoming antique from the modern ML perspective. See Questions for more details.

**Questions:**

1. Could the authors comment on whether TVLARS would be useful in datasets that are large in modern conception---ImageNet, large NLP datasets, etc?
2. I didn't understand what Figure 2 was trying to say. Is the point of Figure 2 to show that warmup is better than no-warmup in some way?

---

> ### Author Response · Authors · 2023-11-20
> **Response to Reviewer k26Y**
>
> 1. **Comments on whether TVLARS would be useful in datasets that are large in modern concept:** We thank the reviewer for a very constructive comment.
> - We agree that challenging datasets are required to verify a good algorithm. However, due to the short rebuttal phase, it is impossible for us to generate the results according to the challenging dataset such as ImageNet for the revised paper. We will implement and add the results according to the ImageNet in our public repository in the future.
> Besides, we want to discuss that the TinyImageNet can be considered as a challenging dataset for top AI conferences. Despite its volume compared to ImageNet, TinyImageNet is still considered as a challenging dataset and widely used in various research which has been accepted in top AI conferences (e.g., CVPR, ICCV, ICLR, ICML, NeurIPS). For instance:
>    - [Stochastic Marginal Likelihood Gradients using Neural Tangent Kernels (ICML 2023)](https://proceedings.mlr.press/v202/immer23b.html)
>    - [Diffusion Model as Representation Learner (ICCV 2023)](https://openaccess.thecvf.com/content/ICCV2023/html/Yang_Diffusion_Model_as_Representation_Learner_ICCV_2023_paper.html)
>    - [Online Continual Learning through Mutual Information Maximization (ICML 2022)](https://proceedings.mlr.press/v162/guo22g.html)
>    - [Delving into Out-of-Distribution Detection with Vision-Language Representations (Neural IPS 2022)](https://proceedings.neurips.cc/paper_files/paper/2022/hash/e43a33994a28f746dcfd53eb51ed3c2d-Abstract-Conference.html)
>    - [The Heterogeneity Hypothesis: Finding Layer-Wise Differentiated Network Architectures (CVPR 2021)](https://doi.org/10.1109/CVPR46437.2021.00218)
>
> - Moreover, in various LBL research, the authors use less challenging dataset to evaluate the LBL performance (e.g., Cifar100, Cifar10, TIMIT, MNIST). For instance:
>   - [On Large Batch Training for Deep Learning: Generalization Gap and Sharp Minima (ICLR 2017)](https://openreview.net/pdf?id=H1oyRlYgg)
>
> - In addition, TinyImageNet is a challenging classification task that it contains 200 categories with 500 samples for each class. However, each sample's size is 64 $\times$ 64, which makes the model harder to extract the useful features from the image. Moreover, inheriting from ImageNet, the images are sampled in various scenarios, but with a much lower number of training samples, the model trained by using this data set will face heavily the out of distribution and long-tailed learning problem as discussed in the above papers.
>
> 2. **Why we should drop the warm-up:** We the reviewer for the helpful comment. We acknowledge that the main problem is the mutual information between Figures 1 and 2. Furthermore, we want to discuss that we use Figure 2 to show the relationship between $\Vert w \Vert$, $\Vert \nabla w\Vert$, and the test loss in different scenarios. By doing so, we can go to the conclusion that the stable convergence behavior in LBL come along with some characteristics (e.g., the stable reduction in $\Vert w \Vert$, and $\Vert \nabla w\Vert$) (mentioned in Section 3.2).
>
> - Per your comment, we have removed Figure 1 (as it is already shown in Appendix H).

---

> > ### Comment · Reviewer_k26Y · 2023-11-21
> > **Response to Rebuttal**
> >
> > I sincerely thank the authors for their hard work in producing the rebuttals and answering my questions. I am afraid I am not able to offer any constructive comments at this point. I will keep my score, and leave it for the area chair to decide whether this paper is an interesting contribution.

---

### Official Review · Reviewer_Kfjf · 2023-10-31

**Soundness:** 4 excellent
**Presentation:** 3 good
**Contribution:** 4 excellent
**Rating:** 6
**Confidence:** 4

**Summary:**

The authors conduct a very thorough empirical study of existing SOTA per layer learning rate adaptation rules, LARS and LAMB. They then identify key patterns in what makes or breaks a good training trajectory and relate that back to shortcomings in the per layer adaptation rules. They then propose key changes that should reduce these shortcomings, and they package them in TVLARS.

Their proposed method is then empirically evaluated and ablated over a variety of datasets, learning rates, batch sizes and tasks. TVLARS appears to have significantly better generalization performance than its competitors, and improved training stability.

**Strengths:**

- Very thorough empircal evaluation of existing methods is already very valuable
- Empirical evaluation of existing methods from new angles adds further value
- The authors identify key shortcomings in their empirical observations that align with theoretical understanding.
- The authors then combine theoretical understanding and their empirical insights to craft a new method, TVLARS, resulting in a well rounded motivation for the method.
- The new method is thoroughly evaluated
- The new method appears to have a significant gain margin over the baselines.

**Weaknesses:**

- The writing throughout suffers from clumsiness, flow problems and structure problems.
- The key method description and presentation is hidden in a number of paragraphs, and the key algorithm box is hard to parse.
- The authors often assume that their audience should know certain things that are not likely to be known by the average deep learning research scientist.

**Questions:**

Suggestions:

Please take the time to rewrite key parts of the paper, for example, in the abstract:

For example, some constructive comments for the abstract:

Your abstract on LBL optimization introduces pivotal concepts and promising methodologies. I offer succinct feedback for refinement:

- Clarify "sharp minimizer" to aid reader comprehension.
- Detail the theoretical gaps in warm-up techniques, emphasizing the contribution of your work.
- Articulate the distinguishing features of TVLARS to highlight its novelty.
- Quantify "competitive results" to underscore the empirical strength of TVLARS.

These focused enhancements will sharpen the abstract's precision and academic rigor.

Then, in your methodology section:

- Clearly introduce the ingredients of your method.
- Provide intuitive explanations for each
- Cross reference these explanations in your algorithm diagram.
- Try to reduce the parsing complexity of your algorithm -- or even better use a nice figure that showcases how your method compares to existing methods functionally and mathematically.

---

> ### Author Response · Authors · 2023-11-20
> **Response to Reviewer Kfjf**
>
> We thank the reviewer for their time and efforts, as well as their valuable comments.
>
> As per the reviewer's suggestions, we have made the following revisions to our paper:
>
> We have revised the abstract along with the introduction in order to:
> - We provided references to sharp minimizers along with a brief definition for the reader's comprehension.
> - We have explained briefly the potential issues of warm-up techniques when incorporating them into LARS and LAMB to tell why we have to propose our new algorithm TVLARS.
>
> We have revised the empirical analysis (section 2) and methodology (section 3) in order to:
> - We highlighted the difference between our TVLARS and WALARS to address your comments about the clarification of the paper’s motivation and contributions.
> - We conducted further empirical experiments according to layer-wise learning rate annealed by warmup strategy to give a more thorough understanding about the shortcomings of using warm-up in LARS and LAMB.
> - We demonstrated the algorithm more systematically and clearly defined the algorithm’s ingredients.
> - We revised and improved the paper consistency in using notations and abbreviations.

---

### Official Review · Reviewer_XbTS · 2023-11-04

**Soundness:** 3 good
**Presentation:** 2 fair
**Contribution:** 2 fair
**Rating:** 5
**Confidence:** 3

**Summary:**

This paper focused on the convergence stability of large-batch training. More specifically, the authors noticed that recent methods, such as warmup, lack theoretical foundation and therefore they try to conduct empirical experiments on LARS and LAMB. Based on the above analysis, they propose a  novel algorithm Time Varying LARS (TVLARS) to make the initial training phrase stable without the need of warm-up. The experimental results on CIFAR-10, CIFAR-100 and Tiny-ImageNet also illustrates that the proposed method can further improve the performance of LARS and LAMB for large-batch training.

**Strengths:**

1. This paper tries to drop warmup and make large-batch training stable is very interesting. Since we usually use a large learning rate when batch size is very large and warm-up is important to make the training process stable.
2. The proposed method is very easy to follow.

**Weaknesses:**

1. Although this problem is very interesting, but I still not very clear why we should drop warmup. In my experience with large-batch training, I think warmup is a very simple and important method.
2. For your proposed method, I think we need to tune more hyper-parameters to get a great performance, which may make the proposed method less convenient to use.
3. Although the proposed method can improve the performance of LARS and LAMB when we don't use warm-up, I noticed that the accuracy is still too low compared with LRAS/LAMB + warm-up.

**Questions:**

1. I think layer-wise optimization methods (LARS/LAMB) are very sensitive to the initialization methods of weights. For eq. (3), the layer-wise learning rate depends on the weight norm of each layer since the gradient is normalized. Therefore, I think the reason why the initial training process is unstable is related to the initialization method. So my question is whether you try to use different initialization methods and analyze their results.

---

> ### Author Response · Authors · 2023-11-20
> **Response to Reviewer XbTS**
>
> 1. **Why we should drop the warm-up:** We totally agree that warmup is a very simple and important method while our proposed method requires parameter tuning to get great performance. However, we want to discuss that:
> - The existing warm-up strategy comprises two phases: 1) a gradual rise in the scaling ratio of the learning rate from 0 to $\frac{B}{B_\textrm{base}}$, and 2) a gradual decay in the scaling ratio of the learning rate from $\frac{B}{B_\textrm{base}}$ back to the lower threshold $\gamma_\textrm{min}$. From these work flow, we have two observations:
>    - First of all, we recognize that phase 1) is redundant. When the scaling ratio $\gamma^k_t$ is excessively small, especially in the early stages, the learning process fails to prevent the memorization of noisy data [[1]](https://arxiv.org/abs/1908.01878), [[2]](https://openreview.net/forum?id=H1oyRlYgg). Additionally, if the model becomes trapped in the sharp minimizers during the warm-up phase, the steepness of these minimizers prevents the model from escaping and further loss landscape exploration. As a consequence, this process consumes a significant amount of computation resources and learning time.
>    - Secondly, as the decay ratio of the warmup technique in LARS is fixed by $\frac{1}{2}\Big[1 + \cos\Big(\frac{t-de}{T-de}\pi\Big) \Big]$, where t is the current time step and the T is the total timestep. Therefore, the decay characteristics are the same among all datasets and model architectures. As a result, when applying to different datasets (with different sharp minimizers distributions [[3]](https://papers.nips.cc/paper_files/paper/2018/hash/a41b3bb3e6b050b6c9067c67f663b915-Abstract.html)) and model architectures, the learning is not adaptable and tunable to achieve the best performance.
>
> - As per your comment, we have:
>   - Revised the introduction, LARS analysis, and proposed method to indicate the drawbacks of warmup in LARS and emphasize our TVLARS motivations and contributions.
>   - We conducted further empirical experiments according to the layer-wise learning rate value to reinforce our observations about the redundant learning phase when incorporating warm-up into LARS and LAMB. The experiments are demonstrated in Appendix J.
>
>
> 2. **Hyper-parameter tuning needs to be tuned more which is less convenient to use:**
> In our proposed method, there are several parameters that do not need tuning. However, to conduct a fair comparison between TVLARS, LARS, and LAMB, they are fixed at specific values.
> - In TVLARS, the layer-wise learning rate $\gamma_t^k$ is annealed by the formula: $\phi_t = (\alpha + e^{\psi_t})^{-1} + \gamma_{\min}$ where $\psi_t = \lambda(t - d_{\rm e})$. A fair comparison is conducted when the target learning rate of the three optimizers and the final lower threshold $\gamma_{\rm min}$ are the same, which means $\alpha$ is always set to $1$ and $\gamma_{\rm min}$ is set to $\frac{\mathcal{B}}{\mathcal{B}_{\rm base}} \times 0.001$.
> - As a result, only one parameter needs tuning: $\lambda$. As we have stated above, the use of $\lambda$ is crucial as the decay characteristics may not be the same among datasets and model architectures. Therefore, using $\lambda$ can not only make the optimizer adaptive to different scenarios but also to different sizes of batch, as we made an ablation test on this in Section 5.2.1.
> - Finally, using those parameters, upper and lower boundaries are constructed to make the training more stable.
>
> 3. **The accuracy is still too low according to LARS/LAMB:**
> - As we can see from Table 1, our TVLARS has higher accuracy in almost every case of classification tasks. Moreover, in the cases that our proposed method did not dominate the WALARS, our performance gap between WALARS is trivial (i.e., $<0.3\%$).
> Especially, in SSL tasks, our TVLARS outperforms LARS and LAMB in every case. Notably, in many cases, we achieve from 5 to 10% higher than LARS and LAMB in terms of accuracy.
> - Moreover, as we mentioned in answer to questions Q1, and Q2, our TVLARS method does not only outperform warmup LARS and LAMB in terms of accuracy but also achieves faster convergence. This significant improvement comes from the cancellation of the redundant ratio scaling phase (as mentioned in Figure 1), which makes the statistical model stuck at the sharp minimizer.

---

> ### Author Response · Authors · 2023-11-20
> **Response to Reviewer XbTS (second part)**
>
> 4. **Try to use different initialization methods and analyze their results.** We thank the reviewer for the constructive comment.
> - We agree that the layer-wise optimization method is sensitive to the initialization method of weights. In response to your comments, we have carried out further comprehensive experiments under different model initialization methods and the results have been added in the revised manuscript. As we can see from the extensive ablation test, our proposed TVLARS significantly outperforms the two baselines. Besides, we want to discuss more about our implementation and first manuscript. We want to reveal why we can claim that our proposed method is carefully considered in terms of fine-tuning.
> - First of all, we claim that we do use the same method of weight initialization as the authors of the LARS and LAMB method, which is the Xavier Uniform sampling method [LARS](https://github.com/borisgin/nvcaffe), [LAMB](https://github.com/tensorflow/addons/blob/master/tensorflow_addons/optimizers/lamb.py). However, taking this challenge as a chance, we conducted more experiments with different methods of weight initialization including [Xavier Uniform, Xavier Normal](https://proceedings.mlr.press/v9/glorot10a.html), [Kaiming He Uniform and Kaiming He Normal](https://openaccess.thecvf.com/content_iccv_2015/html/He_Delving_Deep_into_ICCV_2015_paper.html), which are the most popular methods nowadays. After conducting those experiments, we found out that the methods of weight initialization do not affect the performance of the training model in LBL scenarios as the results are close to the first ones.
> - Among all weight initialization methods, the method that we used is known to be the most popular and widely used ere are some official implementations that use LARS optimizer with the same weight initialization method as our implementation.
>   - [BarlowTwins](https://github.com/facebookresearch/barlowtwins/tree/main)
>   - [VICReg](https://github.com/facebookresearch/vicreg)
>   - [SimCLR](https://github.com/google-research/simclr)

---

### Author Response · Authors · 2023-11-20
**Revision of Paper**

Dear all,

We would like to announce that we made a major revision to our submission, according to the reviewers' valuable comments. We have marked major revised sentences and paragraphs in blue. We keep other minor revisions in terms of coherence and notation consistency in black.

Some noteworthy changes include:
- In the abstract and introduction:
  - We provided references to sharp minimizers along with a brief definition for the reader's comprehension.
  - We have explained briefly the potential issues of warm-up techniques when incorporating them into LARS and LAMB to tell why we have to propose our new algorithm TVLARS.

- In the empirical analysis (section 2) and methodology (section 3):
  - We highlighted the difference between our TVLARS and WALARS to address your comments about the clarification of the paper’s motivation and contributions.
  - We conducted further empirical experiments according to layer-wise learning rate annealed by warmup strategy to give a more thorough understanding about the shortcomings of using warm-up in LARS and LAMB.
  - We demonstrated the algorithm more systematically and clearly defined the algorithm’s ingredients.
  - We revised and improved the paper consistency in using notations and abbreviations.

- In the experimental evaluation:
  - We revised the evaluation in order to concise our writing.
  - We conducted further ablation tests to evaluate our proposed algorithm on various model initialization methods to reveal that our proposed algorithm outperforms other baselines in all settings.

- In Coding Supplementary Material, we also provided the option argument to switch among methods of weight initialization for reviews to evaluate conveniently.

We also plan to add the following to the camera-ready version, if the paper gets accepted:
- Implement additional experiments on more challenging data, such as ImageNet.
- Provide up-to-date coverage of existing results, including a summary table.

Thanks,

Authors

---

### Meta-Review · Area_Chair_xLSD · 2023-12-11

**Metareview:**

This paper studies training stability of LARS. In particular, the paper examines the necessity of warm up for stable training of LARS and potential issues due to warmup. Furthermore, the authors propose an algorithm Time Varying LARS (TVLARS), which enables stable initial phase training without the need of warm-up. The authors provide experimental evidence to support their claims.

**Justification For Why Not Higher Score:**

I agree with the reviewers that the motivation, presentation & rigor of the paper need to be improved significantly before acceptance at a conference. For these reasons, I recommend rejection in the current form.

**Justification For Why Not Lower Score:**

N/A

---

### Decision · Program_Chairs · 2024-01-16

Reject